# What Was Old Is New Again: The Pennate Diatom *Haslea ostrearia* (Gaillon) Simonsen in the Multi-Omic Age

**DOI:** 10.3390/md20040234

**Published:** 2022-03-29

**Authors:** Noujoud Gabed, Frédéric Verret, Aurélie Peticca, Igor Kryvoruchko, Romain Gastineau, Orlane Bosson, Julie Séveno, Olga Davidovich, Nikolai Davidovich, Andrzej Witkowski, Jon Bent Kristoffersen, Amel Benali, Efstathia Ioannou, Aikaterini Koutsaviti, Vassilios Roussis, Hélène Gâteau, Suliya Phimmaha, Vincent Leignel, Myriam Badawi, Feriel Khiar, Nellie Francezon, Mostefa Fodil, Pamela Pasetto, Jean-Luc Mouget

**Affiliations:** 1Institute of Marine Biology, Biotechnology and Aquaculture, Hellenic Centre for Marine Research (HCMR), Gournes Pediados, 71003 Heraklion, Greece; gabednoujoud@yahoo.fr (N.G.); jonbent@hcmr.gr (J.B.K.); khitembenali24@gmail.com (A.B.); 2Oran High School of Biological Sciences (ESSBO), Cellular and Molecular Biology Department, Oran 31000, Algeria; 3Laboratoire d’Aquaculture et Bioremediation AquaBior, Université d’Oran 1, Oran 31000, Algeria; 4Laboratoire Biologie des Organismes, Stress, Santé, Environnement (BiOSSE), Le Mans Université, Avenue Olivier Messiaen, 72085 Le Mans, France; aurelie.peticca@univ-lemans.fr (A.P.); orlane.bosson@univ-lemans.fr (O.B.); julie.seveno@univ-lemans.fr (J.S.); helene.gateau@univ-lemans.fr (H.G.); suliya.phimmaha.etu@univ-lemans.fr (S.P.); vincent.leignel@univ-lemans.fr (V.L.); myriam.badawi@univ-lemans.fr (M.B.); feriel.khiar@univ-lemans.fr (F.K.); mostefa.fodil@univ-lemans.fr (M.F.); jean-luc.mouget@univ-lemans.fr (J.-L.M.); 5Department of Biology, United Arab Emirates University (UAEU), Al Ain P.O. Box 15551, United Arab Emirates; igor.kryvoruchko@uaeu.ac.ae; 6Institute of Marine and Environmental Sciences, University of Szczecin, Mickiewicza 16, 70-383 Szczecin, Poland; romain.gastineau@usz.edu.pl (R.G.); nickolaid@yandex.ru (N.D.); andrzej.witkowski@usz.edu.pl (A.W.); 7Karadag Scientific Station, Natural Reserve of the Russian Academy of Sciences, Kurortnoe, 98188 Feodosiya, Russia; olivdav@mail.ru; 8Laboratoire de Génétique Moléculaire et Cellulaire, Université des Sciences et de la Technologie d’Oran Mohamed BOUDIAF-USTO-MB, BP 1505, El M’naouer, Oran 31000, Algeria; 9Section of Pharmacognosy and Chemistry of Natural Products, Department of Pharmacy, National and Kapodistrian University of Athens, Panepistimiopolis Zografou, 15771 Athens, Greece; eioannou@pharm.uoa.gr (E.I.); kkoutsaviti@pharm.uoa.gr (A.K.); roussis@pharm.uoa.gr (V.R.); 10Institut des Molécules et Matériaux du Mans, UMR CNRS 6283, Le Mans Université, Avenue Olivier Messiaen, 2085 Le Mans, France; nellie.francezon.1@ulaval.ca (N.F.); pamela.pasetto@univ-lemans.fr (P.P.)

**Keywords:** auxosporulation, diatoms, epigenetics, genomics, *Haslea ostrearia*, HBIs, marennine, phylogeny, transcriptome

## Abstract

The marine pennate diatom *Haslea ostrearia* has long been known for its characteristic blue pigment marennine, which is responsible for the greening of invertebrate gills, a natural phenomenon of great importance for the oyster industry. For two centuries, this taxon was considered unique; however, the recent description of a new blue *Haslea* species revealed unsuspected biodiversity. Marennine-like pigments are natural blue dyes that display various biological activities—e.g., antibacterial, antioxidant and antiproliferative—with a great potential for applications in the food, feed, cosmetic and health industries. Regarding fundamental prospects, researchers use model organisms as standards to study cellular and physiological processes in other organisms, and there is a growing and crucial need for more, new and unconventional model organisms to better correspond to the diversity of the tree of life. The present work, thus, advocates for establishing *H. ostrearia* as a new model organism by presenting its pros and cons—i.e., the interesting aspects of this peculiar diatom (representative of benthic-epiphytic phytoplankton, with original behavior and chemodiversity, controlled sexual reproduction, fundamental and applied-oriented importance, reference genome, and transcriptome will soon be available); it will also present the difficulties encountered before this becomes a reality as it is for other diatom models (the genetics of the species in its infancy, the transformation feasibility to be explored, the routine methods needed to cryopreserve strains of interest).

## 1. Introduction

The increase in scientific knowledge in modern biology depends on a few model organisms that represent essential material for fundamental and application-oriented research. Model organisms are extensively used by scientists as references or standards to study cellular and physiological processes in other organisms. This is true when the biology of these models is well understood, usually implying that their genetic variance has been controlled and reduced to a minimum. However, this could restrict their value when the goal is to explore the broad diversity of the tree of life and to extrapolate rules to genetically very different organisms. Hence, in the biological sciences, it is now well recognized that there is a crucial need to investigate new, unconventional model organisms. This especially applies to groups such as microalgae for which biodiversity is high, but also underexplored or underexploited.

In the last decades, there has been a growing interest in microalgae due to their importance in ecology, particularly in ecosystem services, as well as in many possible commercial applications. Indeed, microalgae are responsible for the primary productivity in aquatic ecosystems, thus forming the basis of food webs and accounting for more than 50% of the oxygen produced [1]. Due to a huge bio- and chemodiversity, and possibly high growth performances, they constitute a vast renewable resource [2]. However, only a few strains of microalgae are actually exploited on an industrial scale (less than a dozen, including those used as feed in aquaculture). Consequently, microalgae-derived products are currently limited to a few niche markets; however, these organisms can be a valuable source of proteins, lipids, carbohydrates, pigments, vitamins, and high-value antimicrobial, antiproliferative or antioxidant compounds [3,4,5,6,7].

Aside from the intrinsic fundamental or applied importance of microalgal models, facilitated culture and strain stability can be prerequisites to their development. For diatoms, *Thalassiosira pseudonana* has been the first species sequenced to provide a whole genome [8], which fostered research using multi-omic approaches. Subsequently, more diatom species have had their genomes sequenced, namely: *Phaeodactylum tricornutum* [9] (Bowler et al. 2008), *Thalassiosira oceanica* [10], *Fistulifera solaris* [11], *Cyclotella cryptica* [12], *Fragilariopsis cylindrus* [13], *Pseudo-nitzschia multistriata* [14], *Skeletonema costatum* [15], *Seminavis robusta* [16], *Nitzschia inconspicua* [17] and the non-photosynthetic *Nitzschia* sp. Nitz4 [18]. All these organisms are marine species, either centric (*T. oceanica*, *T. pseudonana*, *S. costatum*) or pennate (*F. cylindrus*, *F. solaris*, *P. multiseries*, *P. multistriata*, *P. tricornutum*, *S. robusta*, *N. inconspicua*, *Nitzchia* sp.). Three of the above-mentioned diatoms are benthic (*S. robusta*, *F. solaris*, *F. cylindrus*), while the rest are planktonic, with the ecological behavior of *P. tricornutum* being as diverse as its morphotypes [19,20]. To date, among these species, sexual reproduction has been evidenced in *C. cryptica* [21], *P. multiseries* [22], *P. multistriata* [23] and *S. robusta* [24].

The microalga *Haslea ostrearia* (Gaillon) Simonsen is a marine pennate diatom that produces the very unique blue pigment marennine (Figure 1). This pigment is water-soluble, and it can be fixed on the gills of many invertebrates, particularly oysters (the so-called ‘green oysters’). The first mention of the species known today as *H. ostrearia* was made in 1820 by François Benjamin Gaillon, a French custom receiver but also a botanist, who studied the cause of the greening of bivalves in northern France [25]. Although using his own limited material, an ‘inferior microscope’ with ‘imperfect optical properties’ [26], Gaillon identified microorganisms contributing to this coloration, to which he gave the name *Vibrio ostrearius*. Soon thereafter, Jean-Baptiste Bory de Saint Vincent proposed a new genus encompassing all microorganisms, with a cell shape like a weaver’s shuttle and acute at the ends—the genus *Navicula* [27]. He designated *Vibrio tripunctatus* Müller as the type species of his new genus, thereafter renamed *Navicula tripunctata*. Accordingly, Bory did the same with *V. ostrearius*, which became *Navicula ostrearia*, the official name of this unique taxon for ca. 150 years. Indeed, in 1974, Simonsen established the genus *Haslea* to accommodate all diatoms with peculiar features of frustule morphology, i.e., lanceolate cells, acute ends, fine transverse and longitudinal valve striations; these are all characteristics ascribed to the section/subgenus *Fusiformes* [28]. Simonsen, thus, renamed *N. ostrearia* as *Haslea ostrearia*, and proposed this taxon as the type species of the genus [29].

In contrast to the two most-studied diatom models, *P. tricornutum* and *T. pseudonana*, *H. ostrearia* is a thychopelagic/benthic organism [30]. Its sexual reproduction is well assessed and controlled in the laboratory [31,32], and it could have an unsuspected impact in natural environments where it blooms [33] (Gastineau et al. 2017). It displays specific metabolic pathways that lead to the production of compounds, such as marennine, haslenes and highly branched isoprenoids (HBIs), which have a high application potential for the food, feed, health and cosmetic industries [33]. One of the main inadequacies of *P. tricornutum* or *T. pseudonana* as models is the absence of known sexual reproduction in these species, which is reflected in the fact that their sizes do not decrease with time. Indeed, due to the rigid frustule being composed of two valves of definite unequal sizes (the larger epivalve and the smaller hypovalve), each daughter cell, after mitosis, inherits only one valve, which becomes the new epivalve. Consequently, in the course of asexual propagation, cells shrink until the original size is restored, due to sexual reproduction. Theoretically, the mechanism and evolutionary path of these species may differ significantly from those in other diatoms, and more models are, thus, needed to better represent the whole taxon.

The phenomenon of greening caused by *H. ostrearia* gives a significant economic added value to oysters produced on the Atlantic coast of France, since green oysters, with a specific flavor and emerald color, are rarer and more expensive than ordinary oysters. Furthermore, it has been shown that in vitro, marennine has biological properties, such as antioxidant, antibacterial and antiviral activities [34,35,36], as well as allelopathic effect against other microalgae [37,38]. Thus, the biological traits of *H. ostrearia* and the possible applications of its unique pigment marennine make this microalga a highly original biological model. The aim of this review is, thus, to advocate the development of this new microalgal model, presenting past achievements, ongoing research, as well as future objectives to be reached before making this diatom a new reference for biology.

## 2. Biodiversity, Chemodiversity, Phylogeny and Phylogeography of the *Haslea* Genus

The species *H. ostrearia* has long been known as the only blue diatom ever described. For decades, any record worldwide of a blue diatom, i.e., a pennate diatom with blue tips, was assigned to *H. ostrearia*. However, recent works on blue *Haslea* strains using scanning electron microscopy (SEM) observation and molecular markers have uncovered unsuspected biodiversity of this taxon, with four new species of blue *Haslea* recently described: *Haslea karadagensis* N. Davidovich, Gastineau and Mouget [36], collected in the Black Sea; *Haslea provincialis* Gastineau, Hansen and Mouget [39], collected in the Mediterranean Sea; *Haslea nusantara* Mouget, Gastineau and Syakti [40], collected in the Java Sea; and *Haslea silbo* Gastineau, Hansen and Mouget [41], from La Gomera (Canary Islands, Spain). In the last decade, our knowledge about the diversity of non-blue *Haslea* has increased as well [42,43,44]. So far, the genus *Haslea* encompasses 44 taxa, of which 36 are taxonomically accepted, as listed in *AlgaeBase* [45]. Most of these species are benthic organisms encountered in the Northern hemisphere, possibly due to a more intense research effort in this part of the globe, or to a higher proportion of land to water, hence the benthic environments.

In a time-calibrated molecular phylogeny [46], *Haslea* belongs to a Naviculoid clade along with *Navicula*, *Hippodonta*, *Seminavis* and *Pseudogomphonema*. Interestingly, *Haslea* diversified as the first of the above clade at the boundary between Cretaceous and Tertiary (ca. 66 my BP), followed by *Hippodonta* with *Navicula* as the last genus [46]. The early divergence of *Haslea* has recently been confirmed [47]. *Haslea antiqua* Fenner has been described based on well-studied specimens (LM and SEM). This new species, found in a deep-sea sediment in the SW Pacific, has apparently been drifted by hazardous events to the deep-sea sediment. Sediment containing *H*. *antiqua* are well dated and represent the Late Paleocene. The specimens illustrated in [47], in terms of morphology, represent characteristics typical of the genus, including the striation, valve layering and particular hasleoid helictoglossae. It is missing the accessory rib on the raphe sternum interior, but evidently possesses a fascia in the valve center, present in some extant species (e.g., *Haslea crucigera* (W. Smith) Simonsen) [48].

Among the 36 recognized taxa, four species (*H. karadagensis*, *H. provincialis*, *H. nusantara* and *H. silbo*) produce blueish compounds, the so-called “marennine-like pigments", in reference to marennine produced by *H. ostrearia*. Furthermore, at least three other species of blue *Haslea* have been discovered, but they have not been formally described yet. Thus, it can be hypothesized that the number of species recognized to produce marennine-like pigments will increase with time, especially if the research effort is sustained. Nevertheless, blue *Haslea* are rarely observed in natural environments, with a few exceptions where they can bloom (see below). Given the scarcity of observations of diatoms, and beyond microalgae with blue cell-compartments, it seems that the ability to synthesize marennine-like pigments is a taxon-specific characteristic that is present in very few taxa, especially with regard to the enormous biodiversity of microalgae.

Several molecular markers (mainly from mitochondrial and chloroplast DNA) have been characterized in *Haslea* species, namely 16S, 18S, COI and *rbcL*. Nevertheless, these markers are not identified or not available for all recently described *Haslea* and for many *Navicula* species. Only the *rbcL* dataset allowed the establishment of a consistent molecular phylogeny comparing different *Haslea* and *Navicula* species. The evolutionary analysis performed using the *rbcL* chloroplast gene (Figure 2) revealed that *Haslea*, as a genus, is monophyletic, being divided into two clades: “blue *Haslea*” and “non-blue *Haslea*”. The phylogenetic analysis also allowed the validation of the inter-genera transfers anteriorly proposed between *Haslea*, *Gyrosigma*, and *Navicula*, particularly concerning *Haslea nipkowii*, *Navicula avium*, *N. howeana* and *N. tsukamotoi* [49,50].

Such a phylogeny based on a single gene is already of great importance for comparative genomic investigations aimed at establishing the molecular classification of *Haslea* spp., and to determine whether the species producing marennine-like pigments emerged from a unique ancestor. However, multigene phylogeny based on several concatenated genes is known to give a better resolution of genetic divergence. Thus, by using annotated genomes in the future, it will be possible to make more accurate phylogenetic trees using concatenated genes, for instance, by detecting the regional association of SNP (significant loci) and linkage disequilibrium plots between the two groups. A much-expected application of the new genomes and the phylogeny-derived closest relatives between blue and non-blue *Haslea* lineages would be to identify the candidate genes involved in the biosynthetic pathway of the marennine-like pigments. Once the candidate genes are known, application-oriented research could be conducted on mutants in the *Haslea* species with the highest capacity of pigment production.

## 3. Next-Generation Sequencing Efforts in *Haslea*

To date, NGS data available for diatom genomics have revealed the complex evolutionary history of diatoms [53]. Previous genomic studies demonstrated that both *T. pseudonana* and *P. tricornutum* genomes are a mosaic of both heterotrophic host and algal endosymbiont genes, as well as apparently non-endosymbiotic, bacterial genes predicted to have been acquired via horizontal gene transfer (HGT) [8,9,54,55]. Six additional diatom genomes, representing both centric and pennate species, have been published [10,11,12,13,15,16,56,57]. Some of the more recently released diatom genomes (e.g., *Cyclotella cryptica*, *Fragilariopsis cylindrus* and *Seminavis robusta*) were generated using long-read sequencing [13,16,57]. Long-read sequencers, such as the Oxford Nanopore Technologies (ONT) MinION device, have the ability to generate ultra-long sequencing reads [58,59] and can yield highly contiguous genome assemblies. However, they are also notoriously error-prone depending on the library preparation method, MinION chemistry and/or base-calling software, with most errors occurring as indels that have the potential to impact downstream analyses, such as gene prediction [60,61,62]. In order to overcome the high per-base error rates associated with MinION sequencing, long-read assemblies are typically corrected or ‘polished’ using ONT raw signal data, as well as high-quality, high-coverage Illumina short-read data [63]. The quality and contiguity of long-read derived genome assemblies can potentially be further improved using optical genome mapping and scaffolding.

### 3.1. Cultivation of H. ostrearia for Genome and Transcriptome Sequencing

The biological material from which a dozen single cells were isolated and respective clonal strains derived (MMS-Nantes Culture Collection, NCC) originated from a wild population of *H. ostrearia* collected in November 2018 in the Bay of Bouin (France). The most actively growing strain—NCC 527, denoted as HoB4 and presumably best adapted to respective local laboratory conditions—was cultivated in two laboratories of the GHaNA consortium: HCMR (Greece) and Le Mans University (LMU, France). It served as the source of genomic data. An F2 medium containing antibiotics was used for HoB4 growth, under 16 h/8 h photoperiod at 19 °C.

As a *Haslea* culture comes with its microbiota, before proceeding to DNA extraction, removing marine bacteria and fungi is essential in order to generate sequencing reads solely originating from diatoms. Several axenization approaches have already been developed for fresh-water and marine species. These procedures include treatment with different antibiotic concoctions on solid media or directly in liquid culture; ultrasonication; filtration through a polycarbonate membrane, following treatment with Triton X-100 and antibiotics; or a combination of methods [64,65,66,67]. In this study, we attempted the removal of contaminating microbial populations from the growing HoB4 cultures using two antibiotics and one antimycotic supplemented as a single solution (10,000 units/mL penicillin, 25 μg/mL amphotericin B and 10 mg/mL streptomycin). Cultures in the mid-exponential phase, treated with antibiotics, were centrifuged at 2500× *g* for 3 min. The pellets were frozen after a short passage of time in liquid nitrogen and stored at −80 °C. The cell pellets were used for DNA and RNA extraction.

### 3.2. gDNA Extraction

The HoB4 genome was sequenced through high-throughput short-read Illumina and long-read ONT sequencing platforms. The combination of Illumina and ONT technologies was chosen to take advantage of short-read (better quality) and long-read (better contiguity) sequencing approaches. One clear application for ONT sequencing is de novo genome assembly, in which long reads facilitate the assembly of long contigs. For the generation of long reads on the ONT platform, the DNA needs to be of high molecular weight, and free of contaminants and nicks.

To achieve this goal, several methods have been tested: different lysis buffers with or without glass beads; different kits; various amounts of the Phenol/Chloroform/Isoamyl alcohol mixture; and RNAse treatment. For some taxa, this is relatively straightforward [51]. However, when experimenting with *Haslea* spp., the presence of metabolites, including marennine, may pose challenges to the isolation of DNA suitable for various molecular applications. When blue strains of genus *Haslea* were cultured under high light intensity, they accumulated more marennine compared with the same strains under low light levels [37]. Genomic DNA from cells grown under strong illumination was less amenable to PCR, presumably due to higher accumulation of photosynthetic products and possibly the marennine itself (Verret et al., unpublished observations). If present during the library construction or sequencing on the ONT platform, this may be expected to negatively impact the results. therefore, we extracted gDNA from *Haslea* cultures grown under low light intensity (50 µmol photon m^−2^ s^−1^) and presenting a reduced accumulation of marennine compared to those grown under standard light intensity (100 µmol photon m^−2^ s^−1^). One widely adopted method of plant DNA isolation entails the inclusion of cetyl trimethyl ammonium bromide (CTAB) in the extraction buffer, which facilitates the separation of carbohydrates from DNA [68]. To enable reliable DNA isolation suitable for genome assembly using the ONT platform for HoB4, we eventually adopted the CTAB-Genomic-tip protocol suggested by ONT for plant leaves [69] with slight modifications, followed by size selection with a Circulomics Short Read Eliminator kit (SKU SS-100-101-01). This final gDNA from HoB4 was used for both ONT and Illumina sequencing. A first aliquot of 7 µg of HoB4 gDNA was used to prepare the ONT library using the Nanopore Ligation Sequencing Kit (SQK-LSK109). The libraries were sequenced on a MinION Mk1B sequencer on a R9.4.1 flow cell, with a total run time of 96 h. The raw data were base-called with Guppy 4.0.11 [70] in the high-accuracy mode, with the minimum quality score set to 7. Illumina paired-end short-read sequencing was conducted by the Norwegian Sequencing Center (Oslo, Norway), using a second aliquot of HoB4 gDNA. The sequencing outputs are presented in Table 1.

While nucleic acid extraction was conducted on cell pellets originating from antibiotic/antimycotic-treated cultures of HoB4, the elimination of contaminating microorganisms was incomplete, as indicated by the presence of bacterial reads in the sequencing results (see below).

### 3.3. RNA Extraction

In order to obtain a reference transcriptome for HoB4 under optimal growth conditions, a volume of 500 mL of each culture was used as a primary material. RNA was extracted using a NucleoZOL and the NucleoSpin RNA Plant and Fungi kit (Macherey-Nagel). First, cells were lysed and homogenized in NucleoZOL reagent. Contaminating molecules such as DNA, polysaccharides and proteins, were precipitated and removed by centrifugation. The NucleoZOL procedure allows for the isolation of both small and large RNA molecules. The final RNA amount was reconstituted in RNase-free water. RNA samples’ integrity was checked by migration on MOPS/Formaldehyde-denaturing agarose gel and on an Agilent Bioanalyzer.

### 3.4. First Insights into the Haslea ostrearia Genome and Transcriptome

ONT sequencing and assembly of the HoB4 genome were carried out at IMBBC-HCMR [71]. ONT sequencing long reads were corrected with the Illumina short reads using Pilon version 1.23 [72]. Both ONT and Illumina data were used to generate several genome assemblies and to compare the performance of different tools and integration strategies, keeping, as a final assembly, the one that obtained the best balance between contiguity, completeness and quality (Figure 3).

After removing six contigs corresponding to six complete bacterial genomes from contaminating organisms (see section about bacteria associated with *Haslea* below), the assembly consisted of 3015 genomic scaffolds (N50 of 121 kb) covering 107 Mb; this is within the range of the estimated genome size of other pennate diatoms [73]. We found a GC content of 45%, which is close to what is generally found in diatoms [18]. BUSCO analysis, using the Eukaryota_odb9 database (100 species) and *Galdieria* ssp. as a reference, showed 266 genes orthologous to genes of other eukaryotic organisms. This number constitutes ca. 88% of the total number of 303 genes found among eukaryotes in the database. A total of 230 of 266 were single-copy genes.

A BLASTn (>95% identity) was executed to search for similarity with the already published *H. ostrearia* strain NCC 396 mitochondrial genome, and the published chloroplast genome of *H. ostrearia* strain NCC 320. The complete chloroplast and mitochondrial genomes were retrieved in this way, each into a single contig.

The transcriptomic data were obtained using Illumina HiSeq sequencing (RNAseq) from HoB4 cultures grown under optimum growth conditions. After quality filtering and trimming, genome-guided de novo assembly of the reads using the Trinity software suite [74], with normalization parameters, showed 64,082,309 total assembled bases, 39,181 total genes and 48,308 total transcripts. The number of genes and transcripts are similar to those detected via de novo assembly of RNA-seq reads with Trinity in the strain NCC 153.8, i.e., 35,813 genes and 46,690 transcripts [75]. At the same time, the number of genes found in HoB4 is somewhat larger than the range of 11,448 to 34,642 genes reported in nine other diatoms [73].

## 4. Bacteria Associated with *Haslea ostrearia*

In a benthic biofilm microenvironment, bacteria and microalgae have close mutualistic, commensal or parasitic interactions [76]. They connect with each other, both having major contributions to the different biogeochemical cycles in the ocean [77,78,79]. These auto- and heterotrophic organisms rely on each other for cross feeding [80], and the molecules mainly exchanged are sources of carbon, nitrogen, vitamins (cobalamin), iron and other trace elements. In natural environments, diatom communities vary depending on abiotic factors—such as temperature, salinity, nutrients, etc. [81,82,83]—and biotic factors, such as competition between diatoms (allelopathy) [37] and the associated bacteria [84]. However, bacteria can also influence diatom communities by interfering with nutrient availability and their reproduction cycle [85,86].

In the case of clonal cultures, the composition of the bacterial community varies according to the strain/species and the physiological state of the culture [87]. The variation of the bacterial community can also be linked to the excretion of exopolymeric substances (EPSs) by diatoms, as EPSs depend on species and environmental conditions [87,88,89].

In our study, we aimed to produce a clonal culture of *H. ostrearia* strain HoB4 to facilitate the completeness and accuracy of the genomic assembly. To this end, we applied various antibacterial and antimycotic agents (see the section on DNA isolation above). Even after these preventive measures, most of the DNA in the samples was of bacterial origin (>70% of Illumina reads). Eight different circular sequences were detected with the 16S region V1-V3, with two of them corresponding to HoB4 mitochondrial and chloroplast genomes. The bacterial sequences were more abundant than the mitochondrial and chloroplast ones, at 79%, 3% and 18%, respectively. The majority of the bacteria detected belong to the branch of *Proteobacteria* (57%), followed by the branch of *Bacteroidetes* (24%) and *Cyanobacteria* (18%) (Figure 4).

Complete genomic sequences were found for the genera *Maribacter* (Flavobacteriaceae), *Mycobacterium* (Mycobacteriaceae), *Paracoccus* (Rhodobacteraceae), *Pseudomonas* (Pseudomonadaceae), and *Sphingorhabdus* (Sphingomonadaceae). Interestingly, genera belonging to Flavobacteriaceae, Pseudomonadaceae, Rhodobacteraceae, and Sphingomonadaceae have already been found to be associated with diatoms [94,95]. This also concerns the genus *Ilumatobacter* (Ilumatobacteraceae) and some cyanobacterial genera, for which we have detected only partial sequences. Furthermore, these taxa are found in sea- or fresh-water, and are commonly found in diatoms’ microbiomes. For the above reasons, these bacteria seem to be tightly associated with *H. ostrearia* strain HoB4, which would explain why they are difficult to remove without killing the diatom partner. Nevertheless, there could be one exception regarding the presence of *Pseudomonas*, as this genus of Gammaproteobacteria is usually found in waste water [96]. The presence of this genus in the sample could result from contamination prior to sample collection. However, it could also be a true satellite of the HoB4 microbiome under natural conditions.

Irrespective of their origin, the presence of bacterial reads/sequences in the sample hampered the assembly of the genome of *H. ostrearia* strain HoB4. Without clean, axenic data and/or previous knowledge about the studied species, it is difficult to identify which sequence belongs to which organism. Their abundance even causes some troubles when rebuilding long sequences of HoB4, by drowning the needed reads in a flood of data. In future projects, efforts will be made to identify the bacteria present in the sample using 16S sequencing on a non-treated sample. With the knowledge of the microbiome associated with *H. ostrearia*, it will be easier to clean the data, or to kill the bacteria with a precise and specific cocktail of antibacterial agents.

## 5. Epigenetics

Epigenetics refers to heritable phenotypic changes resulting from the modification of gene expression, not involving changes in DNA sequence. Epigenetic mechanisms include DNA methylation, histone modification and mi-RNA expression [97,98]. DNA methylation takes place at the fifth position of the cytosine base (5mC) in the CG, CHG and CHH context, with H being any base but C. Methylation allows a genome to respond rapidly and dynamically to environmental changes and is essential for many fundamental biological processes [99]. This modification, which can occur on different bases, is best described on cytosines because cytosine methylation in the CG context is the predominant type of cytosine methylation in higher eukaryotes. In addition, photosynthetic organisms can have cytosine methylation in CHG and CHH contexts, where H is any base but C. Furthermore, apart from methylation of cytosines, methylation of adenine bases is also found in microalgae [100,101], plants, animals and fungi [102]. Methylation is catalyzed by DNA methyltransferases (DNMTs), enzymes which allow the addition of a methyl group to a nucleotide. Up to six families of DNMTs have been described; however, DNMT2s are, in fact, RNA methyltransferases. The main and best known are DNMT1s, which maintain methylation, while DNMT3s induce de novo methylation [103,104].

DNA methylation in diatom genomes has been shown to occur in varying amounts between species. Previous studies have revealed that in the *P. tricornutum* genome, the methylation rate ranges between 2.5 and 6%, and that 5mCs are principally located in repeats (e.g., transposons) but also in some genes [105,106,107]. The variations in methylation positions may depend on the detection method, but also on the culture conditions or age. This rate, however, is much lower compared to the rate of CpG methylation, which is above 70% in somatic cells of vertebrates [99]. Methyltransferase domains are highly conserved throughout the kingdoms of life. However, in diatoms, DNMT families have some specific features. A study performed on the diatoms *P. tricornutum* and *T. pseudonana* showed the presence of DNMTs similar to animal DNMTs (DNMT2-3-5). Nonetheless, the authors reported the absence of DNMT1 genes that would potentially be replaced by DNMT5 genes for methylation maintenance [104,105,106]. In these microorganisms, a specific “bacterial-like” DNMT, whose role is not fully defined yet, has also been found. In addition to these methylation mechanisms, in animals, demethylation is catalyzed by dioxygenases (ten-eleven translocation—TET) [108]. These enzymes catalyze the transition of 5mC to demethylated cytosine through intermediate steps (5hmC, 5fC, 5caC) (Figure 5). In *P. tricornutum*, homologues of these TETs have been found, implying mechanisms similar to those described in metazoans [109]. By revealing gene expression changes, transcriptome analysis can help evidence epigenetic marks.

Regarding epigenetics, research on *H. ostrearia* is at its infancy and has only focused on DNA methylation. A preliminary investigation of *H. ostrearia* transcriptome through OthoFinder [110] and comparison with the *P. tricornutum* methylation gene set has revealed the presence of the whole gene repertoire of DNMTs. The *H. ostrearia* genome includes DNMT3 and 5, as well as the bacterial-like DNMT (Table 2, Appendix A).

The analysis of a whole genome using bisulfite sequencing in *H. ostrearia* has highlighted the presence of methylated cytosine bases within the genome. Indeed, Bismark pipeline analysis shows that 2.67% of cytosine bases are methylated with 12.5%, 0.6% and 0.6% in the CH, CHG and CHH context, respectively. The presence of methylated cytosines was confirmed using ELISA tests, which allowed for determination of the global methylation and demethylation state by quantifying the occurrence of 5mC and 5hmC, respectively, in a DNA sample. In a series of preliminary experiments, an environmentally realistic UVA+UVB irradiation treatment induced demethylation of the *H. ostrearia* genome (5mC) (Figure 6a). Contrarily, the 5hmC test, which allows for monitoring of the presence of an intermediate state of the demethylation process, showed no significant difference between the two conditions (Figure 6b). The same results were obtained after exposure of *H. ostrearia* to the biocide diuron (Figure 6c,d). From these preliminary experiments, it can be concluded that UV and diuron stress induce hypomethylation of the genome in *H. ostrearia*. The observed decrease in DNA methylation could result from a decrease in DNMT activity (passive demethylation) and/or an increase in the activity of demethylases themselves (active demethylation). Regarding the absence of 5hmC in the *H. ostrearia* genome, and more generally in diatom genomes, it could result from the absence of a TET enzyme, allowing the transition from 5hmC to 5mC.

## 6. The Genetic Basis of Sexual Reproduction in *Haslea*

The number of identified representatives of the marine genus *Haslea* continuously increases. Thirty-six taxonomically accepted species are currently listed in the AlgaeBase [45]; however, sexual reproduction has been observed in only five of them, including one non-blue- and four blue-colored ones. It should be mentioned that sexual reproduction and auxosporulation is an obligate stage of the life cycle of the vast majority of diatoms— due to the specific construction of the cell frustule, predetermining its diminution as a result of vegetative divisions—and subsequent restoration of the maximal size during sexual reproduction [111,112].

In unicellular organisms, including diatoms, mating can occur between gametes produced within the same strain (homothallic mating system) or between gametes produced by strains of complementary mating types (heterothallic mating system). Neuville and Daste were the first to describe sexual process in *Haslea* (*Navicula*) *ostrearia* [113,114]. However, that was intraclonal (homothallic) reproduction, and nobody could repeat their experiments for a long time. Thirty-six years later, heterothallic sexual reproduction was described in this species and heterothally was demonstrated as dominant in the breeding system, whereas intraclonal reproduction is rare and sporadic [31].

Heterothallic sexual reproduction was also observed in a second blue *Haslea*, *H. karadagensis*, newly described after its reproductive isolation from *H. ostrearia* was identified [36,115]. Subsequently, the life cycle and sexual reproduction of *H. provincialis* [39] and *H. silbo* [41] were studied. So far, sexual reproduction has been described in only one non-blue *Haslea*: *H. subagnita* [116].

A highly species-specific, finely organized scheme of the auxosporulation process in diatoms requires the manifestation of sexes, which, in most centric species, is regulated epigenetically; meanwhile, in pennates, it is determined genetically and does not change throughout the entire life history of a particular clone. For this reason, most pennate diatoms are diclinous, and cells of two opposite or complementary sexes are needed for sexual process to be triggered [112]. The exceptions are occurrences of paedogamy and automixis.

Until recently it was unclear how the transition from the vegetative phase to sexual reproduction is regulated at the molecular level. The attempt to identify sexual-reproduction-specific genes in diatoms was first undertaken a little more than two decades ago [117]. It was found that multiple genes are upregulated during the onset of sexual reproduction in *Thalassiosira weissflogii* (Grunow) G.A. Fryxell and Hasle, currently regarded as *Conticribra weissflogii* (Grunow) Stachura-Suchoples and D.M. Williams [118]. A novel gene family, referred to as *Sig* genes, has been shown to be expressed during sexual reproduction. Ten sexually inducible genes have been identified, although it was very likely that they represented only a small fraction of all sexual-reproduction-specific genes in *T. weissflogii* [117].

The preliminary stages of gametogenesis may involve complex (in some species) cascade pheromone activity [119,120,121,122], which implies the successive activation of different genes. Evidence for the presence of sex pheromones was provided for several species, all of them being pennate. They demonstrated a predominantly heterothallic mode of reproduction. Multi-stage gene expression presumably allows vegetative cells to exit the mitotic cycle and undergo meiosis to form gametes [117]. Each of the two mating types (hereafter, MT) secretes specific pheromones, triggering MT-specific responses. Interaction via pheromones attracts sexual partners, then stops their cell cycle and synchronizes a switch from mitosis to meiosis and gametogenesis. The pheromones involved in cell cycle arrest are chemically distinct from those used to attract the opposite MTs [121]. Apart from *S. robusta*, involving multiple signal molecules has been demonstrated for *P. multistriata* [121]. A less complex pheromone system was reported for *Cylindrotheca closterium* (Ehrenberg) Reimann and J.C. Lewin, where no evidence of MT+ producing pheromones was found [122].

Until recently, the genetic basis of sex determination was not known for diatoms. Various types of sex determination systems have evolved in different eukaryotic groups [123,124]. For dioecious species, the sex chromosomes are referred to as XY or ZW, depending on which is the heterogametic sex. Sex can also be determined by environmental cues (epigenetic regulation in the case of centric diatoms), by the implication of multiple genes or of cytoplasmic elements. In contrast to epigenetic regulation in centric diatoms, in pennate species, sex factors (designated as M and F) were shown to be inherited independently, and the combinations MF and FF determine male (heterogametic) and female (homogametic) descendants, respectively [125,126,127,128]. The gametes of the investigated species could be differentiated by their behavior and mode of formation, which allows the application of the terms “male” and “female”. Later, the mating type plus (MT+) was confirmed as the heterogametic sex in *S. robusta* [129]. In a study aimed at comparing the expression profiles of the two MTs in *P. multistriata*, five mating-type-related genes were found, of which three were Mating-type Related Plus (MRP) and two were Mating-type Related Minus (MRM). One of the MRP genes seems to be determined by a single mating-type locus [130]. A system based on a single-gene sex determination could potentially have evolved in diatoms. However, more reliable is a system suggesting that two genes are involved, because three genders (males, females and hermaphrodites) may be controlled by a single genetic locus [131,132]. In some pennate diatoms, such as for *H. ostrearia*, bisexual clones also occur, thus revealing the existence of both dioecy and monoecy [133].

Genomic regions associated with sex determination are often characterized by the lack of recombination, which results in reduced efficacy of selection against structural mutations [134]. Theoretically this promotes the creation of chromosomal heteromorphism and determines the size differences between sex chromosomes [132]. It can be noted, however, that dimorphic sex chromosomes are unknown in diatoms. The same is observed in vascular plants, where sex chromosomes have been identified in very few species so far. Among dioecious plants with cytogenetic and/or molecular evidence for the presence of sex chromosomes, male heterogamety (XY) is predominant, while female heterogamety (ZW) is very rare [135]. Shifts between different sex chromosome systems (XY vs. ZW) are also well documented and appear to be a common phenomenon. Sexual systems may differ, even in representatives of the same genus [135]. In some plants, the Y-specific suppressor of female function (SOFF) gene acts as a suppressor of femaleness. Experimental gamma irradiation knockout resulted in the conversion of males to hermaphrodites [136]. Given this case, we can hypothesize that the same type of sex regulation is at work in dioecious, predominantly heterothallic diatoms capable of rare intraclonal reproduction.

Fairly close relatives to diatoms among Stramenopiles, for which information is available regarding life cycles and breeding systems, are the multicellular brown macroalgae. Their sexual dimorphism is expressed either during the diploid phase of the life cycle (in sporophytes) or in haploid male and female gametophytes. Morphological differences between male and female gametophytes in the brown macroalga *Ectocarpus siliculosus* (Dillwyn) Lyngbye are correlated with a few dozen sex-related genes, and in contrast to diatoms, the sex is determined by a UV chromosome system, with a female U and a male V chromosome [137]. Neither the U nor the V chromosomes recombine in the UV system. In diatoms, since no sex chromosomes have been found yet, it is unclear where the genes responsible for sexual differentiation are located on the chromosomes, and whether recombination of these loci is possible.

Regarding the breeding system of *H. ostrearia* and other blue diatoms, both homothallic and heterothallic reproduction types are found in each of these species. However, homothallic reproduction is rare and sporadic, while heterothallic reproduction is dominant and usually abundant [31]. It is interesting to note that all descendants resulting from intraclonal reproduction of *H. ostrearia* for which the mating type has been examined (a dozen strains up until now) turned out to be of the same mating type as their parental clones. This might argue for the homogametic status of *H. ostrearia* clones capable of intraclonal reproduction. Still, more data need to be gathered for a statistically significant conclusion about the status. A reference genome, plus the comparison among genomes of two compatible partners and of a homothallic one, are essential for comprehension of the phenomenon.

## 7. *Haslea ostrearia* as a New Model Species for Marine Ecotoxicology

The pollution of aquatic environments is a very complex issue, resulting from the interaction of chemical and physical factors. This pollution affects all life in these environments by causing damage and functional disturbances in ecosystems. To estimate physiological and molecular modifications induced by these toxins and contaminants, many studies use model species. Diatoms, as photosynthetic organisms, are important bio-sentinels in most aquatic environments. Being the main primary producers [138], they are an essential component in studies that aim to understand the alterations of the whole trophic network. For marine ecotoxicology studies, only a few diatom species (*Asteroplanus karianus*, *Chaetoceros* spp., *Coscinodiscus eccentricus*, *P. tricornutum*, *Skeletonema* spp., *Nitzschia closterium*, *Pseudo-nitzschia* spp., *Thalassiosira* spp.) have been used to estimate the negative effects of pollutants on primary photosynthetic producers. Preliminary bibliographic analysis in PubMed up to 2021 shows that 35.29% of the publications focused on marine toxicology are based on pennate diatoms (64.71% for centric diatoms), but only a few of them are benthic organisms. The diatoms used for these studies are mainly *P. tricornutum*, *Skeletonema* spp. and *Thalassiosira* spp. The genomes of *P. tricornutum*, *S. costatum* and *T. pseudonana* have been sequenced and are available, allowing the identification of many genes involved in cellular mechanisms of detoxification. It has been shown that *H. ostrearia* is a marine diatom with a large geographic distribution, encountered in the Atlantic Ocean, the North Sea, the Indian Ocean, the Mediterranean Sea, and the Pacific Ocean. This microalga is mainly a benthic and epiphytic organism, which sometimes constitutes large biofilms when it blooms on marine sediment or seaweeds, as in Corsica (Mediterranean Sea, Figure 7a,c), in Croatia (Adriatic Sea, Figure 7b), or in North Carolina (western Atlantic Ocean, Figure 7d). Thus, this widely distributed benthic diatom can be considered a representative of subtidal environments.

The recent annotation of the *H. ostrearia* strain HoB4 genome and transcriptome has allowed the identification of several biomarker sequences. Thus, data are now available to characterize the complete coding sequences of multiple markers involved in: the antioxidative responses (Ascorbate Peroxydase, 1197 bp; Catalase, 1581 pb; Fe-Mn SOD, 720 pb; Glutathione Synthase, 1584 bp); the general response to stress (Hsp60, 1740 bp; Hsp70, 1980 bp; Hsp90, 2220 bp); the biotransformation of xenobiotics (Cytochrome P450, 2201 bp; Glutathione Reductase, 1575 bp); and the apoptotic response (metacaspases, 984 and 2124 bp) (Figure 8).

Some housekeeping genes, which may be used as references in qPCR investigations, can also be identified: actin (1134 bp), glyceraldehyde-3-phosphate dehydrogenase (1137 bp), and histone H2A (369 bp). This biomarker set can be used to evaluate the effects of various emerging pollutants in the marine environment (e.g., pesticides, nanoplastics, pharmaceutical compounds) on the gene expression of *H. ostrearia* as a new model species in ecotoxicology.

In vitro, it has been shown that high, environmentally relevant concentrations of marennine, such as in culture supernatants, can display toxicological effects on marine organisms—for instance, delayed embryonic development and mortality in *Mytilus edulis* (mussel) and *Sphaerechinus granularis* (sea urchin), with decreased fertilization success and larval development [139]. Nevertheless, marennine is highly water-soluble, and the concentrations tested could only be observed in vivo during a bloom, when *Haslea* cells form biofilms on the surface of sediments, seaweeds or seagrasses. In these microenvironments, it can be hypothesized that marennine may be at a concentration comparable to or higher than a supernatant, and that its impacts should affect mainly benthic organisms.

## 8. Highly Branched Isoprenoids in *Haslea*

Terpenoids, also known as isoprenoids, are a large and remarkably diverse group of secondary metabolites. With more than 60,000 out of the approx. 300,000 natural products reported to date in the Dictionary of Natural Products, terpenoids represent one of the most often encountered chemical classes [140]. Terpenoids have been exploited throughout human history and are still extensively used as constituents in fragrances, in the food industry as supplements and pigments, and in the pharmaceutical sector [141].

Diatoms produce a variety of terpenoids (Figure 9) including sterols and carotenoids, the latter including fucoxanthin, diadinoxanthin, and diatoxanthin [142,143]. In addition to these well-known terpenoids, a few diatom species belonging to the genera *Rhizosolenia*, *Pleurosigma*, *Navicula*, *Berkeleya* and *Haslea* synthesize highly branched isoprenoids (HBIs) [75,144,145]. HBIs are branched hydrocarbons with 20, 25 or 30 carbon atoms. They can be saturated or unsaturated with one to six double bonds [146]. Even though most HBIs are acyclic, there are a few reports of cyclic derivatives [147,148]. Other functionalized structures of HBIs, such as epoxides and alcohols, have also been reported [149]. HBIs have been found in marine sediments from various locations around the globe. They were first found in marine sediments in Peru, Antarctica, the Gulf of Suez, the North Sea, the north Atlantic Ocean, later in Australia, and in crude oil from Utah, USA; more recently, they have been found in sediments of the western and eastern Mediterranean basins [150,151]. The biosynthesis of HBIs has been shown to take place in different species of the genus *Haslea*, including *H. ostrearia*, *H. crucigera*, *H. pseudostrearia* and *H. saltstonica* [152]. The biogenic source of HBIs was first reported in 1994 by Volkman and co-workers, who identified seven C_25_ and five C_30_ HBIs in laboratory cultures of *H. ostrearia* and *Rhizosolenia setigera*, respectively [144]. A few years later, Belt and co-workers identified three additional C_25_ HBIs with three, four and five double bonds from *H. ostrearia* [153], and new C_25_ and *n*-C_27_ HBIs alkenes in *R. setigera* [154].

### 8.1. Biological Roles and Uses of HBIs

The biological function of HBIs in diatoms is still unknown. In vitro experiments using synthetic HBIs at different pH levels, however, have shown that HBIs can form stable vesicles of low permeability. The position of the double bonds was reported to affect the stability of the vesicles, with the middle C=C bond at C-7 being more suitable for the formation of the vesicles [155]. Despite their unclear biological role, HBIs are extensively used as proxies to reconstruct paleo-sea-ice conditions and establish predictive models [156]. One of those HBI proxies is the monounsaturated C_25_ HBI, called “IP_25_” for ice proxy, with a 25-carbon skeleton (Figure 9c). *H. crucigeroides*, *H. kjellmanii*, *H. spicula* and *P. stuxbergii* var. *rhomboides* were identified as biological sources of IP_25_ in the Canadian arctic after cell isolation from sea-ice samples and GC-MS analysis of their terpenoid contents [157]. One of the main reasons to use HBIs as proxies is their sensitivity to environmental changes. When cultured in large non-axenic batches, *H. ostrearia* produced more HBIs in May compared to June, with HBIs showing a higher degree of unsaturation. The effect of temperature, salinity or other factors on the distribution of HBIs, however, could not be precisely determined [158]. In plants, the level of lipid unsaturation increases in response to the decrease in growth temperature. Indeed, lipids with more *cis* double bonds exhibit a lower melting point, preserving the fluidity and stability of the membrane at low temperatures. In contrast, HBIs possess *trans* double bonds and their degree of unsaturation increases with the increase in growth temperature. In cultures of *H. ostrearia* cultivated at 25 °C, 15 °C and 5 °C, the main HBIs produced were tetra-unsaturated, tri-unsaturated and di-unsaturated (hasladienes) C_25_ alkenes, respectively [158]. In addition to the growth temperature, the growth phase has also been shown to influence HBI production. Indeed, even though HBIs are produced throughout all growth phases, the highest cellular concentration of HBIs was measured in the stationary phase in both *H. ostrearia* and *H. vitrea* [159]. Since nutrient availability is limited in the stationary phase, it has been proposed that HBIs might represent an energy source to survive in unfavorable conditions [160].

### 8.2. Terpenoid and HBI Biosynthetic Pathways

Terpenoid biosynthesis can be divided in three distinct stages, i.e., the early, central and late enzymatic phases. Despite their structural diversity, all terpenoids are derived from the C_5_ isopentenyl pyrophosphate (IPP) unit and its isomer dimethyl allyl pyrophosphate (DMAPP), which are synthetized in the early enzymatic steps [161]. IPP and DMAPP are produced by two distinct pathways: the cytosolic mevalonate pathway (MVA) and the plastidic methylerythritol 4-phosphate pathway (MEP). The MVA pathway is present in archaebacteria and in the cytosol of higher plants, while the MEP pathway is present in eubacteria and in the chloroplast of higher plants [162]. The MVA pathway produces only IPP, while the MEP pathway produces both IPP and DMAPP with an IPP:DMAPP ratio of 85:15 [163]. The IPP:DMAPP ratio is regulated by the enzyme isopentyl diphosphate isomerase (IDI), which catalyzes the interconversion of IPP into DMAPP, marking the starting point of terpenoid biosynthesis in plants [164]. In the central steps, IPP and DMAPP are condensed by chain-elongating *trans* and *cis*-prenyltransferases (PTS) to give rise to prenyl chains of different lengths, i.e., C_10_ mono-, C_15_ sesqui-, C_20_ di-, C_30_ tri-, C_40_ tetra- and C_n_ poly-terpenes [141]. Subsequently, the C_10_ monoterpene geranyl diphosphate (GPP), C_15_ sesquiterpene farnesyl diphosphate (FPP), and C_20_ diterpene geranylgeranyl diphosphate (GGPP) are further condensed to give rise to higher terpenoids (e.g., the C_30_ squalene and the C_40_ phytoene, which are formed by the condensation of two FPP and GPP molecules, respectively). In the final steps, linear terpenoids are subjected to modifications, including cyclization, reduction, isomerization, substitution, and conjugation [165].

Knowledge of the enzymes involved in HBIs biosynthesis in diatoms is still in its infancy. Recent advances in the elucidation of the early and central steps of terpenoid biosynthesis in diatoms have, however, shed some new light on this topic [166,167,168,169]. Diatoms seem to have retained both the MVA and MEP pathways, but there are differences across species in the allocation of the precursors towards the synthesis of different isoprenoids. In *P. tricornutum*, *N. ovalis* and *R. setigera* the final products of the MVA pathway are directed toward the synthesis of sterols in the cytoplasm, while the final products of the MEP pathway are involved in the synthesis of plastidic compounds, such as carotenoids and phytol. In contrast, *H. ostrearia* and the centric diatom *T. pseudonana* use MEP-derived precursors for the synthesis of their sterols [148]. This difference is likely to also apply to HBI biosynthesis. Using isotopic label experiments, Massé and co-workers determined the involvement of the MVA pathway in the biosynthesis of C_25_ and C_30_ HBIs in *R. setigera*, while the MEP pathway appeared as the major route for the biosynthesis of isoprenoids in *H. ostrearia*. This result, however, could be explained by the inability of the diatom to undergo mixotrophic or heterotrophic growth or the existence of a compensation mechanism between the two pathways [170]. The identity of the specific enzymes involved in HBI biosynthesis downstream of the MVA and MEP pathways awaits characterization. The available information indicates that a farnesyl diphosphate synthase (FPPS) is likely to be involved in the synthesis of C_25_ and C_30_ HBIs in *R. setigera* [168]. FPPS is one of the most-studied *trans*-PTSs. It catalyzes the formation of FPP, first by a head-to-tail condensation of DMAPP and IPP, to which a second IPP is added, all in *trans* geometry [171]. FPP serves as a substrate to squalene synthase (SQS), the first enzyme of the sterol biosynthesis pathway, which condenses two molecules of FPP to form squalene [172]. Ferriols and co-workers showed that the overall quantity of HBIs produced by *R. setigera* decreases in a dose-dependent manner when using the FPP inhibitor risedronate, suggesting that C_25_ and C_30_ HBIs are the result of a *head-to-middle* condensation of an FPP and a C_10_- or a C_20_-terpene, respectively [168]. Whether FPPS takes part in the biosynthesis of HBIs in *Haslea* is still under investigation. At the genetic level, a high degree of homology is observed among PTSs found in all domains of life. Protein sequence alignments reveal many conserved domains, among them the First and Second Aspartic Rich Motifs (FARM and SARM). Both domains are responsible for the binding of magnesium ions and are essential for enzymatic activity. In addition to the FARM/SARM, conserved residues located upstream of the FARM have been shown to regulate PTS product chain length [161].

Recent whole transcriptome analysis of the *H. ostrearia* NCC 153.8 strain unveiled the presence of five PTS homologues. Phylogenetic analysis, subcellular localization prediction and functional characterization in the heterologous host expression systems of these PTS homologues provided the means to propose an original model for the central steps of terpenoid biosynthesis in *Haslea* sp. First, an FPPS sharing 57% similarity with the *R. setigera* FPPS was identified and named HoPTS1. This enzyme is probably localized in the endoplasmic reticulum membrane and produces FPP as a precursor to sterols. In addition to PTS1, four polyprenyl diphosphate synthases (HoPTS2-5) with orthologues present in other pennate and centric diatom species were identified. Among them, two were GGPP synthases—one likely to be cytosolic (HoPTS3) and a second one chloroplastic—which are possibly involved in carotenoid biosynthesis (HoPTS5). In addition to PTSs, a phytoene synthase (HoPSY) sharing 72% similarity with *P. tricornutum* PSY was identified and predicted to be located in the chloroplast. Last, an SQS fused to an isoprenyl diphosphate synthase (HoSQS-IDI) was identified and predicted to be located in the cytosol [75]. Homologues of SQS-IDI fusions have been reported in other diatom species, such as *P. tricornutum* and *T. pseudonana*, but also in some haptophyte and heterokontophyte algae, and have been proposed to act as multifunctional enzymes [167,173].

Aiming to assess the suitability of *Haslea* strain HoB4—isolated from the same location as strain NCC 153.8—as a model for the study of HBI biosynthesis in *Haslea* sp., we investigated the possible presence of HBIs in this strain. The *H. ostrearia* strain HoB4 was grown under standard laboratory conditions in non-axenic batch cultures to a total of 6 L. Cells were harvested by centrifugation and submitted to saponification according to Belt et al., 1996 [153]. Subsequently, the presence of HBIs in the non-saponifiable hexane extract was investigated by GC-MS analysis (Figure 10a, unpublished results). Under our culture conditions, the main HBI produced was the C_25_ tetraene (*E*)-2,6,14-trimethyl-10-methylene-9-(3-methylpent-4-en-1-yl)pentadeca-2,6-diene. The structure of this HBI was verified after HPLC purification, ^1^H NMR analysis (Figure 10b) and comparison of its spectroscopic data with those previously reported by Belt et al. from an *H. ostrearia* strain isolated near the coast of England [153]. Thus, strain HoB4 produces HBIs and can be considered suitable to study HBI biosynthesis in *Haslea* sp. Further experiments addressing the possible effects of environmental growth conditions, such as temperature, salinity and nutrient availability, on the quantity and diversity of HBIs produced by NCC527 may provide insights into the genes and functional roles of HBI biosynthesis in *Haslea* diatoms.

### 8.3. Future Directions for the Study of HBI Biosynthesis in H. ostrearia

Despite these recent findings, the specific role of each PTS in HBI biosynthesis is still unknown. Complementary approaches, including omics, reverse genetics and synthetic biology, will be necessary to achieve this objective. Whole-genome sequencing of HBI-producing and non-producing *Haslea* strains and comparative analysis of their PTS repertoires may provide insights into gene candidates for HBI biosynthesis. In addition, targeted mutagenesis of PTS candidates will be invaluable in understanding their enzymatic activity and pinpointing their physiological roles. Various transformation protocols, including biolistic, electroporation and bacterial conjugation, and tools for targeted mutagenesis, such as TALEN and CRISPR-Cas9, are now available in some model diatom species, including *P. tricornutum* and *T. pseudonana* [71]. Such protocols and tools, however, remain to be tested in, and tailored to, non-model diatom species such as *Haslea* spp. In addition, synthetic biology could be harnessed to characterize the activity of PTS candidates. The HBI non-producing model diatom species *P. tricornutum* represents an attractive cell chassis to characterize *Haslea* PTSs since it presents a fully sequenced genome, a well characterized metabolome, and is amenable to transformation [9,174]. Our consortium is currently employing these complementary approaches to characterize the possible role of *Haslea* PTS homologues in the biosynthesis of HBIs.

## 9. Cryopreservation Prerequisite for Algal Transformation Benefit

In the laboratory, it has long been observed that most pennate diatom cultures degenerate with time due to their peculiar mitosis and concomitant size reduction phenomenon (see the section on sexual reproduction). This could become prohibitive if time and money are spent to transform strains and characterize mutants. One solution to guarantee the availability of strains long-term is to cryopreserve cells, a method commonly used for bacteria. However, the cryopreservation protocols and strategy to keep diatom cells alive after freezing are in their infancy (e.g., Stock et al. 2018, and references herein [175]). In particular, diatom cell pellets need to be re-suspended in a cryoprotectant solution prior to freezing. The first attempts to cryopreserve *Haslea* cells were conducted a decade ago, only resulting in survival rates higher than 50% with the addition of a cryopreservation agent and the use of a complex immobilization–dehydration protocol [176]. Thus, finding a better way to cryopreserve *H. ostrearia* is needed, because the efficiency of this process was found to be highly species-specific in pennate diatoms such as *C. closterium*, *Opephora guenter-grassii*, *Pinnularia borealis*, *S. robusta*, and in centric diatoms such as *Cyclotella meneghiniana* and *T. weissflogii* [175].

## 10. Frustule Exploitation

An upscale in microalgae cultures is required to ensure the generation of molecules of interest such as pigments and exopolysaccharides, which have many applications in the food industry as probiotics and natural pigments, or cosmetics and biofuels. Hence, photobioreactors are designed to cultivate microalgae on a large scale in order to obtain important amounts of the added-value compounds.

Diatoms are a major group of microalgae found in oceans, waterways and soils, which exhibit different elegant architectures. Their size varies from a few nanometers to micrometers, presenting hierarchically structured pores; these smart frameworks are extremely difficult or impossible to reproduce with a laboratory synthesis [177]. So far, the interest from academia and industries in microalgae has essentially focused on the organic molecules that can be isolated from microalgae or on the biopolymers that are excreted, while the silica frustules represent waste in the biorefinery approach. For diatom cultures, what is the destiny of the silica frustules after extraction of the interesting molecules? If silica debris are going to be waste, can they be recycled for original applications? Exploratory research was carried out to turn the silica skeleton debris into a useful natural resource, rescuing it and considering it as an inorganic charge, which could be modified to suit targeted applications.

Silica particles are used in many industrial applications concerning our everyday lives, such as fillers in paints and coatings, dental composites, printing inks, creams, lotions and gels, adhesives, and shoe soles, to cite a few. The manufacturing processes for obtaining synthetic silica are mainly divided into wet-chemical hydrolysis and flame hydrolysis. They are generally covered by patents; however, the raw material employed comes directly or indirectly (such as SiCl_4_) from sand or quartz [178]. The arising problem is that there is a global sand shortage worldwide [179] because of its many uses, for instance, to produce glass and concrete. As a consequence, scientists and the public start to realize how negative the economic, socio–political and environmental impact of sand extraction can be. Hence, sand is not an unlimited resource, and the biosilica produced by the diatoms cultivated on a large scale for other purposes could be used for niche applications, offering at least a small contribution to the reduction of sand digging.

On the other hand, elastomers need to be charged to increase their mechanical properties, and a major problem is that of obtaining homogeneous dispersions; commercial hydrophilic silica particles are already used as charges, but they are poorly dispersed in hydrophobic matrices. Therefore, preliminary research was carried out to functionalize the debris of silica frustules from *H. ostrearia* with oligoisoprene brushes obtained from natural rubber. The oligoisoprenes were obtained in a few steps, from the controlled degradation of natural rubber chains and successive chain-end modification, to covalently grafting one extremity to the silica surface [180]. The grafting reactions were firstly carried out and optimized on synthetic silica particles, both dense and mesoporous, and then repeated on silica debris. The thermal and mechanical properties of the hybrid particles were characterized, and the morphology was imaged using SEM and transmission electron microscopy (TEM). As an example of application, composite films were prepared with bare and functionalized silica particles and their dispersion, as well as their physical properties, were studied (Figure 11).

In the literature, there has been an attempt to produce silica particles decorated with polyisoprene chains [181], by polymerizing the isoprene monomer from the silica surface via anionic polymerization. This methodology requires very drastic conditions, such as dry solvents, an argon atmosphere, extremely pure monomers, and sensitive reagents; meanwhile, the novel approach is less demanding in terms of conditions and uses a natural resource instead of a monomer industrially obtained as a by-product of the thermal cracking of naphtha or oil. The two required oligomers with a triethoxysilane group at a chain end and a hydroxyl group at the other (Mn 1900 g/mol, Đ 2.2) and the di-acrylate oligoisoprenes (Mn 4100 g/mol, Đ 2.1) were synthesized from solid natural rubber following a procedure described by Tran et al. [180] (Suppl. Mat. and Met. S1). Model mesoporous silica particles were synthesized, and the TEM images showed that the particles had a perfect spherical morphology, with a 150 nm average diameter, pores oriented perpendicularly to the surface, and 21 Å pore diameter (Figure 12).

The elimination of the organic matter from *H. ostrearia* was carried out by treating the dry biomass (2 g) at RT overnight with a mixture of concentrated nitric and sulfuric acid (30 mL of each), then washing with distilled water until it reached a pH of 7. The frustules were dried at 80 °C and TEM micrographs showed debris with a variety of sizes and morphologies, with complete elimination of the organic content, even from inside the squares of the regular network. BET measures showed that the silica of the frustules was mesoporous too, with 0.35 cm^3^/g pore volume and a 41 Å pore diameter. The grafting conditions were optimized using the mesoporous silica particles (0.5 g) and by incubating them in dry toluene (10 mL) at reflux, overnight, under argon, in suspension, in a 0.09 M solution of silane-oligomers (1.5 g in 10 mL toluene). The same procedure was used with the frustule debris. BET measurements showed that the specific surface for the mesoporous particles diminished from 1,269 m^2^/g to 425 m^2^/g after grafting, while those of the frustules decreased from 336 m^2^/g to 6.4 m^2^/g. TEM images confirmed that both synthetic silica and biosilica were surrounded by a layer of polymer coating (Figure 12). Thick films were obtained by mixing 200 mg of di-acrylate oligomers and a 1 or 5% weight of silica particles, in the presence of a 5% weight of Darocur photoinitiator, and irradiating the formulation deposited in teflon molds. Visual observation showed that both bare synthetic silica and biosilica particles formed aggregates in the film volume, while functionalized particles were more easily dispersed, confirming the initial hypothesis that the rubber thin layer would improve dispersion in the rubber matrix. It is not expected that silica frustules will replace carbon black or silica as charges in elastomers applications, such as joints or tyres; however, this study showed that biosilica can have the same chemistry as the synthetic one, opening the way for other interesting applications.

## 11. Iconic Pigments of Blue *Haslea* Species

### 11.1. Structure and Biological Roles

Many biological activities of marennine have been evidenced in vitro in the laboratory; however, in vivo, the intrinsic importance of this pigment for the diatom itself is still under debate. In a natural environment, huge blooms of *H. ostrearia* or other species of blue diatoms can regularly be observed in the Mediterranean Sea, the Adriatic Sea, or the Atlantic Ocean seashores, with concomitant greening of bivalves; however, the resulting impacts on ecosystems are still to be investigated. This phenomenon is of particular economic interest because blue-colored oysters have a higher market value.

It has been shown that marennine is a water-soluble, possibly phenolic compound, lacking any link to a polypeptide or transition metal and displaying a carbocycle different to that of flavonoids and anthocyanins [182]. The intracellular form of marennine (IMn, 10,751 Da molecular weight; absorbance maxima at 247 and 672 nm) accumulates in acidic vesicular complexes [183], mainly in the apical regions of the cell [184]. The extracellular form of marennine (EMn, 9893 Da molecular weight; absorbance maxima at 247, 322, and 677 nm) exhibits slightly different physicochemical properties to those of IMn, suggesting that marennine undergoes molecular rearrangements during excretion. The complete molecular structure of marennine has not been fully elucidated and little is known about the biological role of marennine for the *Haslea* cells, except that it is apparently not directly involved in photosynthesis [185].

Recent research on the chemical structure of marennine has led to the investigation of its electrochemical behavior. Indeed, marennine color changes with pH, the addition of reducing or oxidizing agents, and potential tension were studied using complementary techniques [184]. Furthermore, its chromophore has been shown to be unaffected by the acidic and basic or mechanical treatments used for its extraction. Although marennine color changes with pH and a redox environment, its blue–green shade remains stable in the presence of common food antioxidants and preservatives, such as ascorbic acid and sodium sulfite. Therefore, marennine appears to be a good candidate for a new food colorant, especially in acidic preparations [186].

### 11.2. Applications in Human Health

Preliminary studies using *H. ostrearia* aqueous extracts containing marennine displayed antiviral, anticoagulant [187] or human-cell-growth-inhibiting properties [188]. These activities were confirmed later using purified marennine, which exhibited antibacterial, antiviral, and antiproliferative activities [189]. In this study, both EMn and IMn inhibited the development of marine bacteria, particularly the pathogenic organism *Vibrio aestuarianus*, at concentrations as low as 1 μg/mL. This anti-vibrio effect was dose-, strain- and species-dependent [34]. EMn and IMn, however, did not display any effect on a wide range of pathogenic bacteria that are relevant for food safety. Both forms of the pigment also exhibited antiviral activity against the HSV1 herpes virus. Moreover, they were effective in inhibiting the proliferation of various human cancer cell lines (M113 melanoma cells; SKOV3 and SHIN3 ovarian cancer cells; SW116 colon cancer cells; R3111 kidney cancer cells; 1355 lung cancer cells; and MCF7 breast cancer cells) [189].

### 11.3. Applications in the Food Industry

The production of marennine from *H. ostrearia* is known to whoever has bathed in seas containing the living microalga, as the water becomes blue during the bloom period. The spontaneous release of marennine by the microorganisms makes this natural pigment extremely interesting for industrial applications, as normally, long extraction procedures using organic solvents and various separation techniques are required to isolate dyes from plants or flowers, in addition to the generation of organic wastes from the unused parts of the vegetables. The colorimetric properties of the external form of marennine have been characterized [182]: the UV–visible spectra showed two absorption peaks in the UV region (maximum wavelength at 247 and 322 nm at pH 8) and one absorption peak at 677 nm in the visible region, at pH 8. It was shown that the maximum light absorption wavelength changes reversibly with pH, causing a shift from a blue color in acidic pH to a green color in basic pH [182,190]. Colorimetric analyses were conducted under the International Commission on Illumination protocol [191] using standard light sources corresponding to different environments, in order to have an objective color measurement in view of possible applications in the food or cosmetic industries [190].

Looking for possible applications in the food packaging field, an experiment was conducted to see if the change in the blue–green color could be induced by the contact of marennine with biogenic amines that are produced by fish decomposition. Several solutions of biogenic amines were prepared at a 1 mg/mL concentration (1000 ppm): histamine, cadaverine, putrescine (10 mg in 10 mL H_2_O); tyramine (10 mg in 1 mL DMF + 9 mL H_2_O); tryptamine (10 mg in 2 mL ethanol + 8 mL H_2_O); dimethylamine (30 µL of 40 w% solution + 12 mL H_2_O); and trimethylamine (50 µL of 24 w% solution + 12 mL H_2_O). Two negative controls were prepared (1-octen-3-ol (10 mg in 10 mL ethanol) and hexanal (10 mg in 10 mL H_2_O)) and two positive ones (bromophenol blue (10 mg in 10 mL ethanol) and bromocresol green (10 mg in 10 mL ethanol)). A stock solution of purified marennine was prepared in a phosphate buffer at pH 8; then, other solutions at pH 2, 4 and 6 were prepared, adjusting the pH by the addition of drops of a concentrated HCl solution. 80 µL of each marennine solution at different pH levels were added into a 96-wells plate, as well as 80 µL of bromophenol blue, bromocresol green, water, and a marennine solution in ethanol. 20 µL of each biogenic amine solution was added in each well, as well as the same volume of distilled water, ethanol, DMF/water 90/10, and the controls 1-octen-3-ol and hexanal. As expected, the addition of all the biogenic amines induced a clear shift of color of the bromophenol blue and bromocresol green, from yellow to deep blue; meanwhile, the addition of distilled water, ethanol, DMF/water 90/10, and the controls did not change the yellow color, proving that the experiment was validated. A change in color from blue to green was only observed upon the addition of tyramine, dimethylamine and trimethylamine to the marennine solution in ethanol. In a second, similar experiment, the results observed with marennine in ethanol solution were confirmed; moreover, marennine aqueous solutions taken at different steps of the purification procedure were tested, and a similar shift from the violet color at acidic pH to a green color was observed, by adding tyramine, dimethylamine, and trimethylamine, as well as putrescine and cadaverine. UV–visible spectra were recorded in another experiment, and tyramine, dimethylamine and trimethylamine again induced a shift of color in marennine ethanolic solutions. The initial spectrum of marennine in ethanol showed two maximum wavelength absorption peaks at 620 and 668 nm; upon adding trimethylamine, the peak at 620 nm disappeared, while the one at 668 nm shifted at 700 nm (Figure 13). As the change in color could be perceived by the naked eye when marennine was in presence of certain biogenic amines, work is ongoing to complete this study and use marennine to detect food degradation.

The antioxidant properties of pure marennine were also studied using two different tests in vitro: antioxidant capacity assays (β-carotene and thymidine protection assays and an iron-reducing-power assay) and free-radical scavenging assays (DPPH·, O_2_^·−^, and HO^·^) [35]. In this work, marennine exhibited significantly higher antioxidative and free radical scavenging activities than ascorbic acid, a natural antioxidant commonly present in food. Thus, marennine represents an interesting blue compound for the food industry due to its solubility in water and the intensity of its blue color. However, the evaluation of the long-term stability and of the possible toxicity or mutagenic properties of marennine represent an important step for its use as a food additive. Nevertheless, human consumption of green oysters for centuries without any record of disease or anaphylactic reactions suggests the non-toxic nature of this blue pigment in the context of regular food consumption.

As the preliminary tests showed that marennine also changed color depending on the oxidation state, an in-depth study of its redox properties was carried out by adding reducing and oxidizing agents in aqueous solutions and monitoring the changes using different techniques [184], such as UV–Vis and Raman spectroscopy. Three oxidation states were identified: a first initial or native state that corresponded to the oxidized form (characterized by the standard potential EMnox); a second intermediate one (more blue, EMni); and a third reduced, yellow state (EMnred). The values of the corresponding potentials were estimated (E° vs. ENH, 25 °C) for the EMnox/EMni couple between +1780 mV (H_2_O_2_/H_2_O) and +815 mV (O_2_/H_2_O) for a pH below pKa; between +815 mV (O_2_/H_2_O) and +80 mV (dehydroascorbate/ascorbic acid) for a pH above pKa; and for the EMni/EMnred couple between +80 and −660/740 mV. The current hypothesis on marennine structure (based mainly on ^1^H-NMR evidence) is that it is a polysaccharide linked to a small chromophore, which is responsible for the absorption in the UV–Vis region. Ongoing studies demonstrate that acidic hydrolysis simplifies the ^1^H-NMR spectrum of marennine, and monosaccharides have been found in the colorless supernatant after the reaction; meanwhile, the blue-colored organic phase that is supposed to contain the chromophore shows interesting new signals. This blue fraction was also subjected to cyclic voltammetry; after acidic hydrolysis, the aqueous reaction solution was extracted with THF, and the blue organic phase was purified in a SPE cartridge and eluted with methanol. For the electrochemical measurements, the blue methanolic solution was added in a 0.1 M phosphate buffer solution adjusted to pH 4 using HCl. KCl 0.1 M was added to the PBS solution to maintain the ionic strength constant. A 3-electrode cell configuration was used. The working electrode was a glassy carbon electrode from Bioanalytical Systems Inc., West Lafayette, Indiana, USA (model MF-2012; 3 mm in diameter) and the counter electrode was a platinum wire. All potential values were referred to the SCE system. A bipotentiostat from CH Instruments (model 920C) was used. A profile similar to the one obtained for the marennine solution was obtained, with a reversible redox system that exchanged electrons and protons (Figure 14).

### 11.4. Application in Biosensors

The possibility of using marennine as a sensing agent in a redox sensor was considered in view of practical applications, and an experiment was conducted to measure the amount of oxygen needed to turn the yellow reduced form into the blue oxidized form. A quartz cell containing 1 mL of purified marennine solution (corresponding roughly to 2 mg of marennine), in a phosphate buffer at pH 7, was put in a UV–Vis spectrophotometer, under stirring. The detector of oxygen connected to the oximeter was placed just in front of the cell, sealed with a rubber septum. The marennine solution was initially green and 20 µL of a10 mM dithionite aqueous solution was added; the solution turned yellow and it was flushed with argon to stabilize this state. The oxygen was then let in, and the amount needed for re-oxidation was measured (Table 3).

The marennine solution turned green again, indicating that the change in color is detectable, even when working with small concentrations of marennine and small volumes of oxygen. Further investigation is needed to optimize the system, but these first results show that marennine has the characteristics required to be the sensing element in a redox sensor, including the reversibility of the change in color.

### 11.5. Application in the Cosmetic Industry

In addition to potential applications of a natural blue pigment in the food industry, better knowledge of the properties of marennine and marennine-like pigments could allow their use in the cosmetic industry; however, this would require a large scale culture of the microalga. Indeed, for several years, *H. ostrearia* was grown industrially under indoor, controlled conditions to produce marennine; this was mainly used for the intensive greening of oysters (transfer of knowledge from University of Nantes to the Société de Production de Micro-Algues, SOPROMA, Bouin, France), but also to support cosmetic applications (e.g., Topiderm, Badische Anilin und Soda Fabrik BASF), Florham Park, New Jersey, USA). The SOPROMA production ceased ca. 15 years ago for economic reasons, and the sustainable mass production of this peculiar diatom is still a challenge. Regarding applications in cosmetology, the only test of purified marennine ever published concerns anti-inflammatory activity [33], with the application of a cream containing 10% (m/m) of marennine on mouse ear edema. Applying the cream allowed a reduction of the edema by 62.5%, a moderate effect compared to that of the control topical corticosteroid used for the study (100% inhibition, at 0.1% m/m). Despite the low potential for valorization of marennine in soothing creams, more studies should be conducted, as the demand for algae-derived compounds still exists in cosmetology.

## 12. Conclusions

Phylogenetic studies indicate that *Haslea* species that produce the blue pigment marennine form a well-defined clade inside the genus *Haslea*. This finding sparks great interest in characterizing their gene repertoire and genome evolution. Marennine presents many biological activities, with possible applications in the cosmetic, food and health industries. Thus, we propose the diatom species *H. ostrearia* as new and original model organism for fundamental studies to the phycological community; it also represents an exciting source of new bioactive compounds with high economic potential for many industrial applications.

## Figures and Tables

**Figure 1 marinedrugs-20-00234-f001:**
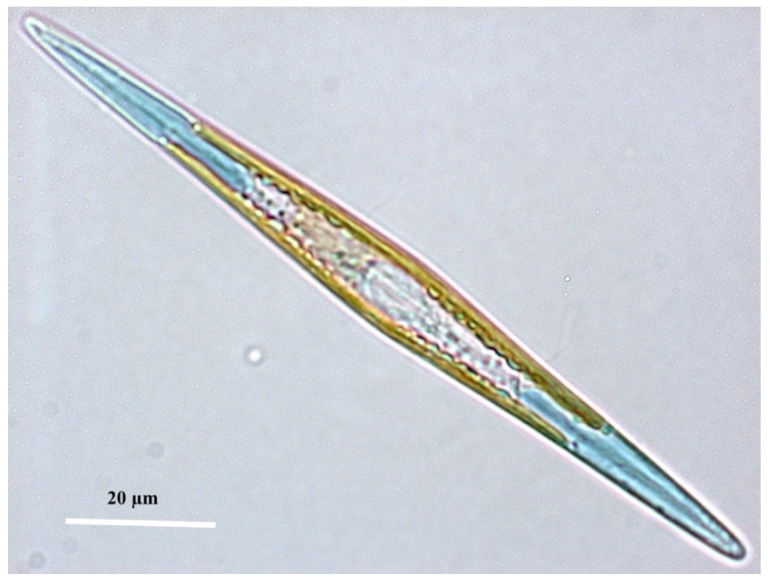
Living cell of *Haslea ostrearia* (NCC 527) observed in light microscopy.

**Figure 2 marinedrugs-20-00234-f002:**
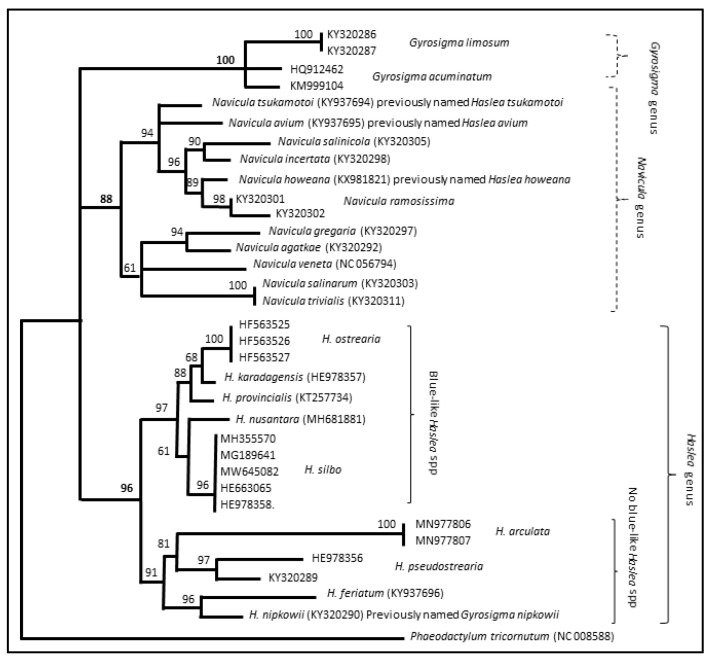
Molecular phylogeny of 22 species from genera *Haslea*, *Navicula* and *Gyrosigma* based on rbcL gene (1065 bp). The evolutionary history was inferred using the Maximum Likelihood method and General Time Reversible model (test of phylogeny: bootstrap, 1000). Multiple alignment of sequences was carried out using MAFFT [51]. Phylogeny analysis was conducted in PHYML [52].

**Figure 3 marinedrugs-20-00234-f003:**
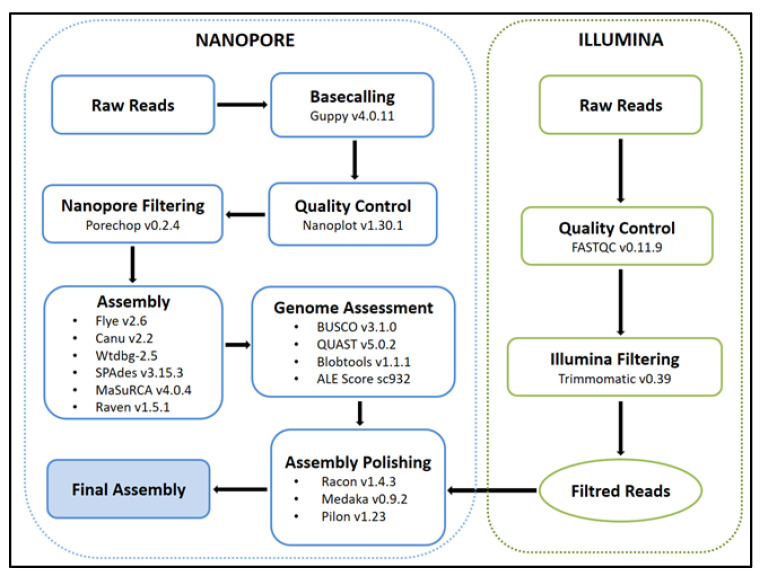
Flowchart of the bioinformatic pipeline used to assemble the HoB4 genome.

**Figure 4 marinedrugs-20-00234-f004:**
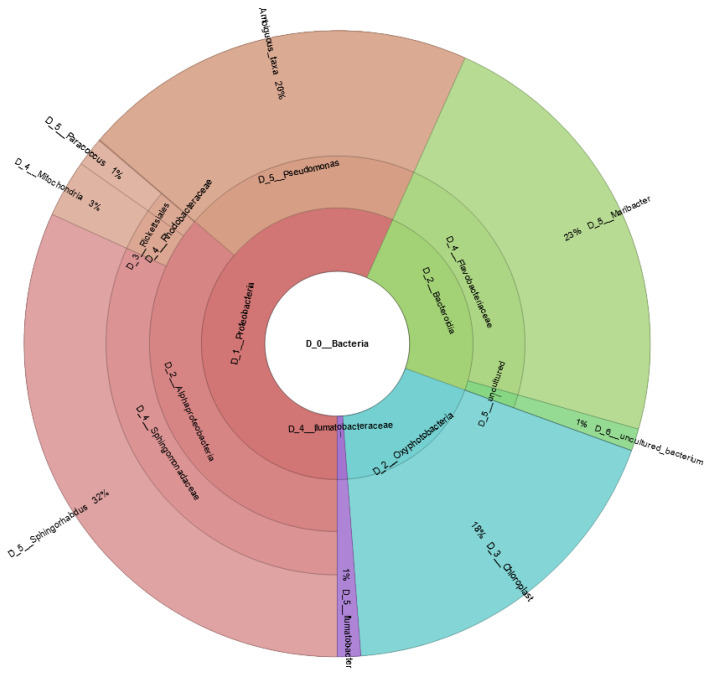
Krona plot of the microbiome detected by searching 16S amplicons in HoB4 metagenomic assembly. The plot displays relative abundance and hierarchy with the use of a radial space-filling display. The amplicons were searched with primers targeting the V1-V3 region of the 16S rRNA gene [90]. The taxonomy was assigned using Qiime2 2021.2.0 and the Silva database 132_99_16S [91,92] and the krona plot was made using KronaTools 2.7.1 [93]. The relative abundance was estimated using the mean depth associated with the bacterial contigs.

**Figure 5 marinedrugs-20-00234-f005:**
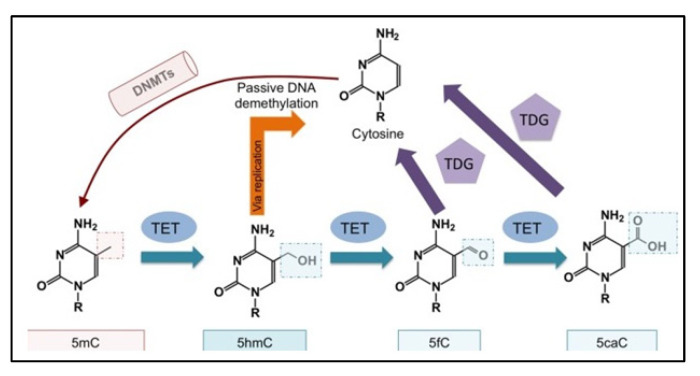
Methylation pathway by DNMTs (DNA methyltransferases) and demethylation by TET (ten-eleven translocation) proteins. The latter will oxidize 5mC to 5hmC (5-hydromethylcytosine). Several oxidations via the TET proteins lead to the formation of 5fC (5-formylcytosine) and 5 caC (5-carboxylcytosine). TDG (thymine DNA glycosylase) allows the excision of these last two bases and the return to an unmethylated cytosine. Figure adapted from [103].

**Figure 6 marinedrugs-20-00234-f006:**
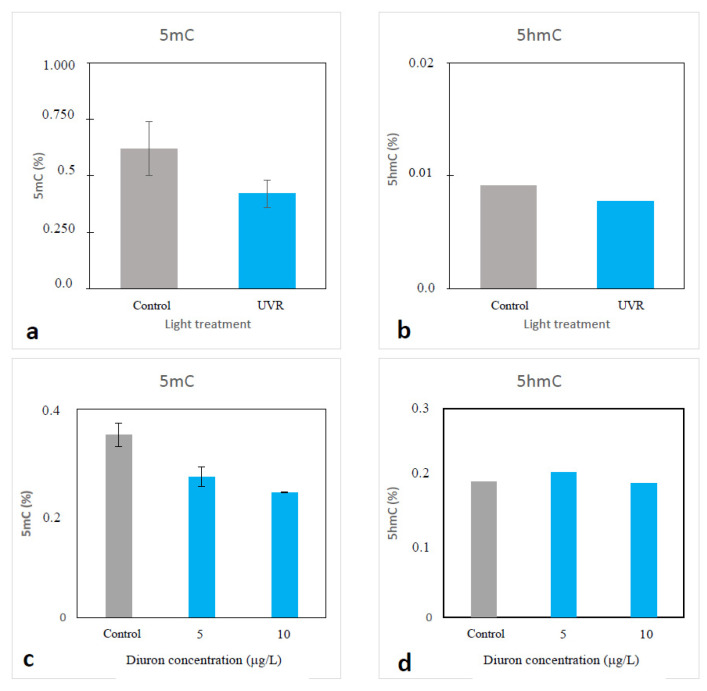
Results of ELISA tests performed using *H. ostrearia* exposed to (**a**,**b**) ultraviolet (UV) stress or (**c**,**d**) diuron treatments. In both conditions, cells were grown under 100 µmol photons m^−2^ s^−1^, and under 14 h/10 h L/D cycles. The 2 h UV radiation (UVR) stress consisted of UVB (450 mW m^−2^) + UVA (11 W m^−2^) exposition. Diuron was used at different concentrations (0; 5 and 10 µg/L). Graphs 5mC and 5hmC correspond to the ELISA assay quantifying the global level of C-methylation and active demethylation of the genome, respectively.

**Figure 7 marinedrugs-20-00234-f007:**
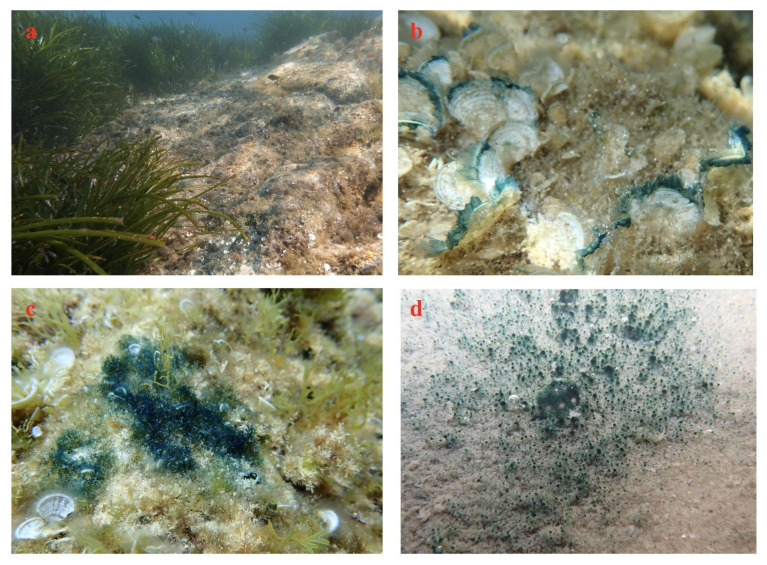
In situ blue *Haslea* species blooms, observed in natural environments in (**a**,**c**) the Mediterranean Sea (Calvi, Corsica, France), (**b**) the Adriatic Sea (Dubrovnick, Croatia, B), and (**d**) the Atlantic Ocean (Morehead City, NC, USA).

**Figure 8 marinedrugs-20-00234-f008:**
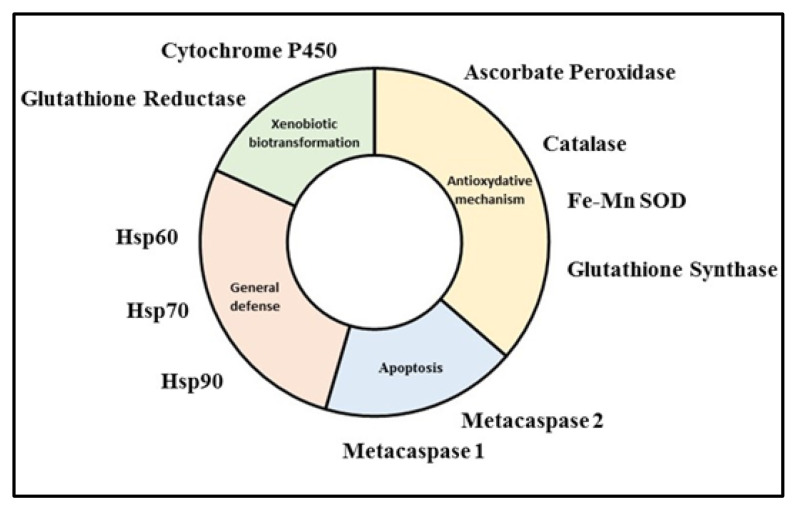
Molecular biomarkers involved in the defense of cellular mechanisms characterized in *Haslea ostrearia* HoB4.

**Figure 9 marinedrugs-20-00234-f009:**
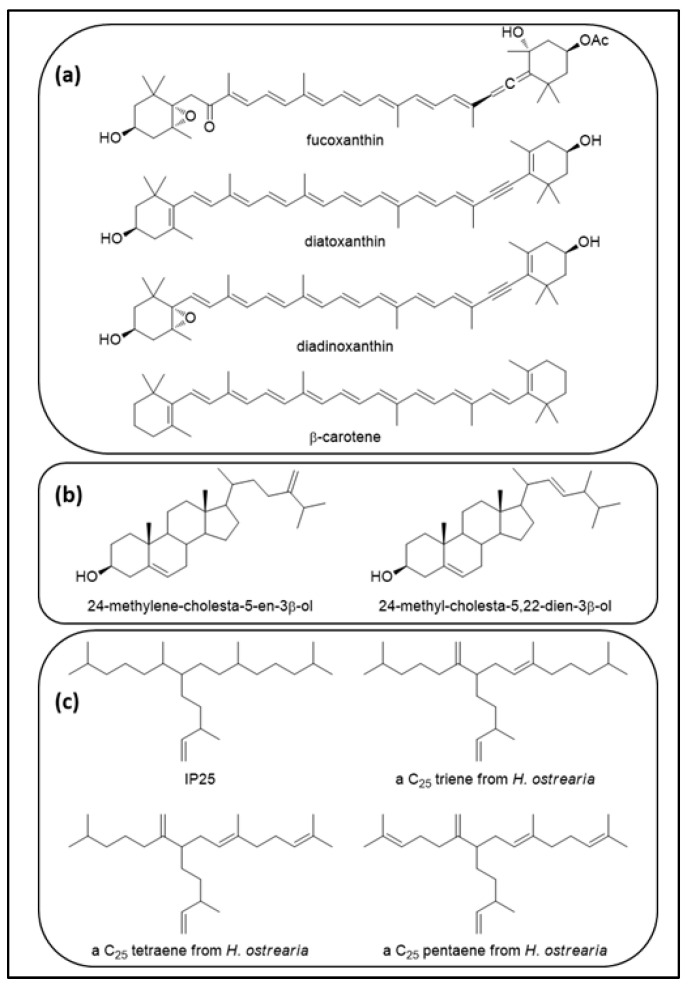
Structure of different terpenoids produced by diatoms: (**a**) carotenoids; (**b**) sterols; and (**c**) highly branched isoprenoids. Figure adapted from [143].

**Figure 10 marinedrugs-20-00234-f010:**
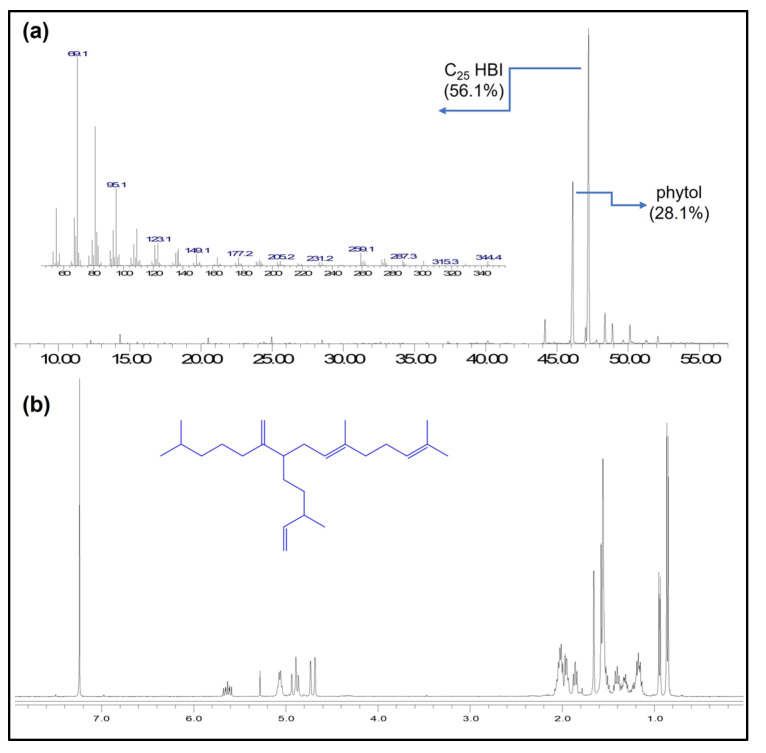
(**a**) GC-MS chromatogram of the non-saponifiable hexane extract of *Haslea* HoB4 strain and mass spectrum of the major HBI detected; (**b**) ^1^H NMR spectrum of the C_25_ tetraene (*E*)-2,6,14-trimethyl-10-methylene-9-(3-methylpent-4-en-1-yl)pentadeca-2,6-diene isolated after HPLC purification.

**Figure 11 marinedrugs-20-00234-f011:**
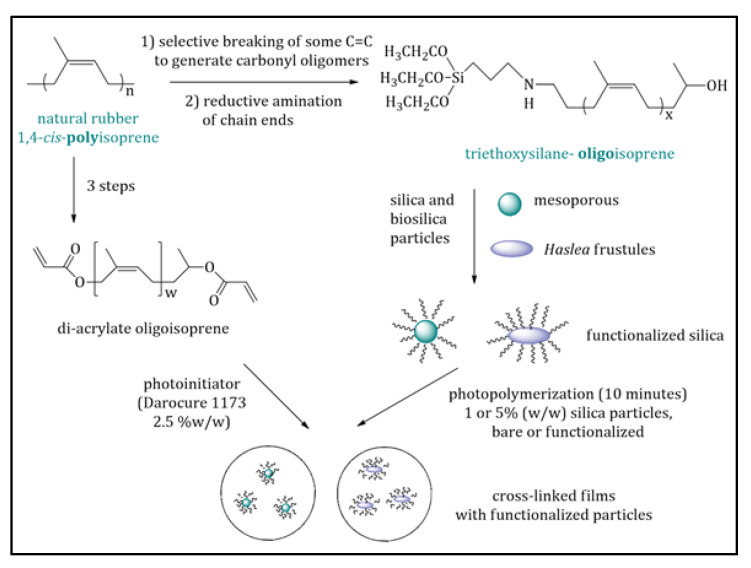
Generation of functional oligomers from natural rubber, grafting on model silica particles and on frustule surfaces, and preparation of charged elastic films.

**Figure 12 marinedrugs-20-00234-f012:**
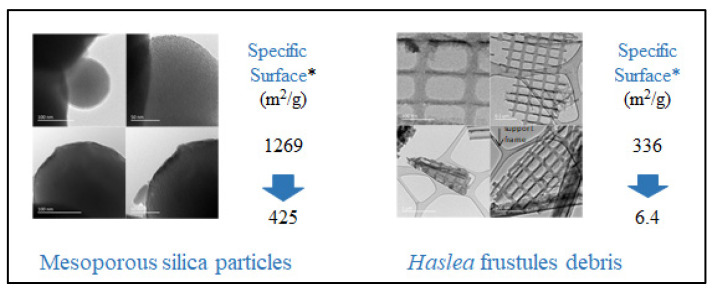
Mesoporous silica particles: (**top**) *before* grafting triethoxysilane oligoisoprene; (**bottom**) after grating triethoxysilane oligoisoprene. *Haslea* frustules: (**top**) after elimination of organic matter; (**bottom**) after grafting triethoxysilane oligoisoprene. TEM micrographs (from JEOL JEM 2100HR, LaB6, 200 KV£). * Specific surface from BET measurements.

**Figure 13 marinedrugs-20-00234-f013:**
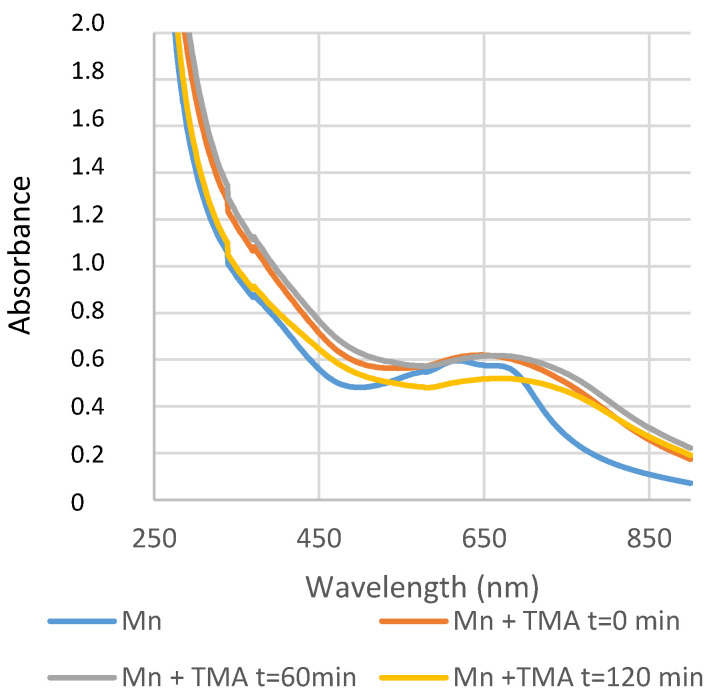
UV–Vis spectra of marennine (Mn) and the effect of trimethylamine (TMA) on its color, monitored for 120 min.

**Figure 14 marinedrugs-20-00234-f014:**
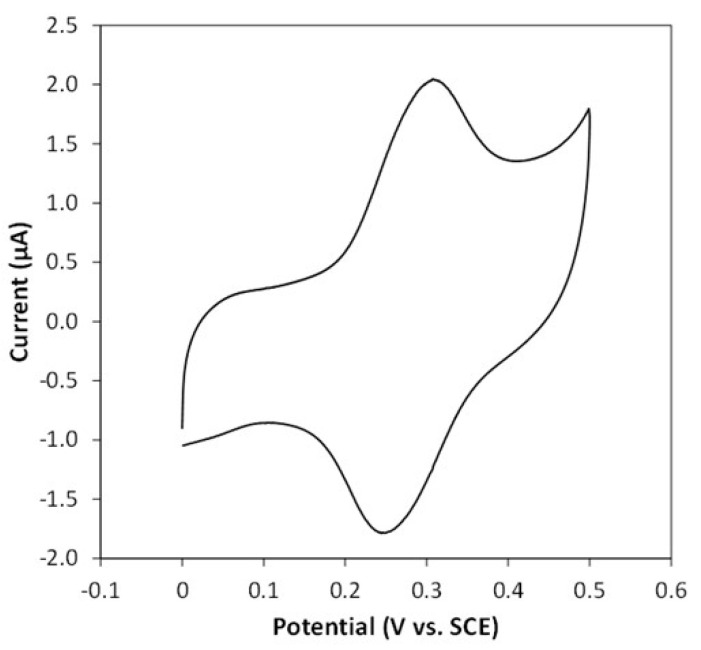
Cyclic voltammetry recorded at 100 mV·s^−1^ on a polished glassy-carbon electrode in a phosphate buffer solution (pH 4) containing the marennine acidolysis product (0.5 mg·mL^−1^).

**Table 1 marinedrugs-20-00234-t001:** Summary of HoB4 genomic sequencing reads produced in this study.

Read Type	No. of Reads	No. of Bases
Illumina	2 × 25,693,372	7,708,011,600
ONT	2,598,470	22,685,729,920

**Table 2 marinedrugs-20-00234-t002:** DNMT homologues identified in HoB4.

Gene Family/Subfamily	ID Phaeo	InterProScan Description
DNMT3	Phatr3_J46156	DNA (cytosine-5)-methyltransferase 3A
DNMT5	Phatr3_EG02369	C-5 cytosine methyltransferase
Bacterial-like DNMT	Phatr3_J47357	C-5 cytosine methyltransferase
Demethylase	Phatr3_J48620	

**Table 3 marinedrugs-20-00234-t003:** Oximetry test: amount of oxygen required to bring back marennine to the blue color after reduction with dithionite (yellow color).

V_dithionite_ (µL)	A_700 nm_	A_610 nm_	A_650 nm_	O_2_ (mg/mL)	A_700 nm_	A_610 nm_
20	0.801	0.748	0.712	0.089	0.844	0.830
40	0.698	0.616	0.715	0.087	0.845	0.833
80	0.448	0.377	0.400	0.072	0.851	0.840

## Data Availability

The raw sequence reads used in this study to generate the HoB4 genome, methylome and transcriptome are available on request from the corresponding author.

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
