# Peer review of "What Was Old Is New Again: The Pennate Diatom Haslea ostrearia (Gaillon) Simonsen in the Multi-Omic Age"

_marinedrugs, 2022, doi:10.3390/md20040234_

Round 1

Reviewer 1 Report

The review is very well conceived and organised, treates all important aspects related with the biology, ecology and biotechnology of species included in the genus Haslea, focusing in the interest of using these organisms as models. I cannot find anymore to add.

Author Response

None needed

Reviewer 2 Report

Dear Editor,

The manuscript entitled “What was old is new again: the pennate diatom Haslea ostrearia (Gaillon) Simonsen at the multi-omic age” describes recent studies performed related to bioinformatics, biochemistry, phylogeny, reproduction and biodiversity of mainly blue Haslea genus. Potential usage areas of this genus in biotechnology were addressed such as in cosmetics, as natural silica nano-particle source, natural colorant, antioxidants, anticoagulant, antibacterial, antiviral agents, as biosensor for oxygen presence etc. Detailed information about the terpenoids found in Haslea were given and they were compared with the ones present in other diatom species. The study presents a good summary of investigations performed related to Haslea genus.

I suggest the publication of the manuscript with minor revision.

Please also consider the questions and suggestions under the sections below;

3.2. gDNA extraction

On Line 272: Genomic DNA from cells grown under strong illumination was less amenable to PCR, presumably due to higher accumulation of photosynthetic products and possibly the marennine itself (Verret et al., unpublished observations).

So does this mean DNA samples were extracted from brown cells which were grown under low light?

  1. Bacteria associated with Haslea ostrearia

Even after the application of antibiotic and antimycotic agents there were many bacteria inside of cultures in the study. This seems as an inevitable case. However, blue Haslea should have less bacterial flora than brown ones. It was suggested in the manuscript that in the future 16S DNA sequences will be performed for untreated samples to determine other bacterial types. Maybe it can be good to do this with the blue cultures as well, if possible, to understand better which ones disappear with marennine.

7. Haslea ostrearia as a new model species for marine ecotoxicology

Authors may also mention about some negative effects of marennine on invertebrates (Falaise et al. (2019). Harmful or harmless: Biological effects of marennine on marine organisms. Aquatic Toxicology, 209, 13-25)

Reviewer 3 Report

Authors present an extensive argument for adoption of H. ostrearia as a diatom model organism. This is a very comprehensive (183 references!) and clearly written review. The strongest argument I see is the fact that this species reproduces sexually, whereas other diatom model organisms reproduce asexually. The history, genetics, epigenetics, ecology, and secondary metabolites of H. ostrearia are well covered in this review article. I think it will appeal to a broad audience. 

Minor points:

Line 412: has been shown to "occur"...

Line 798: "This could become redhibitory if..." Not sure what redhibitory means here.

Line 812: This paragraph, lines 812-852, is too long. Please break it down to smaller paragraphs.

Reviewer 4 Report

This is a very interesting and complete review work about marine pennate
diatom Haslea ostrearia. Still there are some information that should be clarified and revised. Some remarks are given to authors improve the work.

- In the main results should be stated.

- Line 73-76: references are missing.

- Last sentence of introduction should be deleted (ln 140-143).

-
Ln 160: Complete the sentence: In a molecular XX
- Ln 200-203: revise the sentence -> and determine if, and if yes...

- Figure 4: it is hard to read the figure. The size of the letter should be
improved.

- The quality of figure 14 is too low, must be improved.

- Conclusion: are these preliminary results? Authors should revise the
information based on the literature cited.

- References: The name of the species and genus should be written in
italic.
